# Palmitoylation regulates neuropilin-2 localization and function in cortical neurons and conveys specificity to semaphorin signaling via palmitoyl acyltransferases

**Eleftheria Koropouli[1†], Qiang Wang[1‡], Rebeca Mejías[2], Randal Hand[1§], Tao Wang[3], David D Ginty[4], Alex L Kolodkin[1]***

[1]The Solomon H Snyder Department of Neuroscience, Johns Hopkins University School of Medicine, Baltimore, United States; [2]Department of Physiology,University of Seville, Seville, Spain; [3]McKusick-Nathans Institute of Genetic Medicine and Department of Pediatrics, The Johns Hopkins University School of Medicine, Baltimore, United States; [4]Department of Neurobiology, Howard Hughes Medical Institute, Harvard Medical School, Boston, United States

*For correspondence:
kolodkin@jhmi.edu

Present address: [†]First Department of Neurology, Aiginition Hospital, National and Kapodistrian University of Athens School of Medicine, Athens, Greece; [‡]Department of Neurology, Zhongshan Hospital, Fudan University, Shanghai, China; [§]Prilenia Therapeutics Development LTD, Herzliya, Israel

**Abstract** Secreted semaphorin 3F (Sema3F) and semaphorin 3A (Sema3A) exhibit remarkably distinct effects on deep layer excitatory cortical pyramidal neurons; Sema3F mediates dendritic spine pruning, whereas Sema3A promotes the elaboration of basal dendrites. Sema3F and Sema3A signal through distinct holoreceptors that include neuropilin-2 (Nrp2)/plexinA3 (PlexA3) and neuropilin-1 (Nrp1)/PlexA4, respectively. We find that Nrp2 and Nrp1 are S-palmitoylated in cortical neurons and that palmitoylation of select Nrp2 cysteines is required for its proper subcellular localization, cell surface clustering, and also for Sema3F/Nrp2-dependent dendritic spine pruning in cortical neurons, both in vitro and in vivo. Moreover, we show that the palmitoyl acyltransferase ZDHHC15 is required for Nrp2 palmitoylation and Sema3F/Nrp2-dependent dendritic spine pruning, but it is dispensable for Nrp1 palmitoylation and Sema3A/Nrp1-dependent basal dendritic elaboration. Therefore, palmitoyl acyltransferase-substrate specificity is essential for establishing compartmentalized neuronal structure and functional responses to extrinsic guidance cues.

## Editor's evaluation

Signaling mediated by Semaphorins and their receptors Nrp1 and Nrp2 is crucial for regulating the morphology of dendritic spines and dendritic arborization during development. In this manuscript, the authors found that the post-translational modification of S-palmitoylation dictates the subcellular localization and trafficking of Nrp2, but not Nrp1, and is required for Sema3F-dependent pruning of spines on the apical dendrites of layer V cortical neurons. The study provides important insights into how semaphorin signaling achieves spatial specificity on diverse downstream cellular events.

## Introduction

The central nervous system (CNS) consists of numerous disparate classes of neurons with remarkably distinct morphologies and functions. This is particularly prominent in laminated structures, including the cerebral cortex, where pyramidal neurons occupying different cortical layers acquire distinct

morphologies and their processes exhibit subcellular compartmentalization that mediates distinct functions (*Spruston, 2008*). For example, dendritic spines, which receive the vast majority of excitatory inputs, have specific subcellular distributions that directly impact electrical properties of neurons and subsequently the activity within neuronal circuits. Numerous studies highlight the physiological importance of dendritic spines, linking alterations in spine number and morphology to various neuropsychiatric disorders (*Penzes et al., 2011*).

One class of proteins implicated in nervous system development are semaphorins (*Koropouli and Kolodkin, 2014*). Class 3 secreted semaphorins semaphorin 3F (Sema3F) and semaphorin 3A (Sema3A) play critical roles in several aspects of neural development and function, including axon guidance, axon pruning, dendritic arborization, dendritic spine distribution, synaptic transmission, and homeostatic synaptic plasticity (*Danelon et al., 2020*; *Demyanenko et al., 2014*; *Gu et al., 2003*; *Koropouli and Kolodkin, 2014*; *Li et al., 2022*; *Riccomagno et al., 2012*; *Sahay et al., 2005*; *Tran et al., 2009*; *Wang et al., 2017*). In mammalian cortical neurons, Sema3F mediates pruning of excess dendritic spines along the apical dendrite of layer V cortical pyramidal neurons, whereas Sema3A promotes the elaboration of basal dendrites in the same neurons (*Tran et al., 2009*). Sema3F and Sema3A bind distinct holoreceptor complexes that include neuropilin (Nrp) and plexin (Plex) transmembrane proteins; Sema3F exerts many of its effects via a holoreceptor complex that includes Nrp2/PlexA3, whereas Sema3A acts through a holoreceptor complex that includes Nrp1/PlexA4 (*Yaron et al., 2005*). Recent work provides insight into the signaling pathways that mediate Sema3F/Nrp2-dependent cytoskeletal rearrangements resulting in dendritic spine pruning in cortical neurons, including contributions by specific immunoglobin superfamily transmembrane proteins that mediate select responses to secreted semaphorins and proteins that regulate actin cytoskeleton dynamics (*Demyanenko et al., 2014*; *Duncan et al., 2021*). However, molecular mechanisms that regulate neuropilin subcellular localization and underlie divergent Sema3F and Sema3A functions in cortical neurons remain largely unknown.

The attachment of the fatty acid palmitate on thiol groups of cysteine residues, known as S-palmitoylation (hereafter referred to as palmitoylation), is a reversible posttranslational modification that dynamically regulates protein localization and function of a vast protein repertoire in different tissues (*Linder and Deschenes, 2007*; *Salaun et al., 2010*). In the nervous system, palmitoylation is critically involved in all aspects of neural development and function (*Fukata and Fukata, 2010*; *Kang et al., 2008*), including axon outgrowth (*Tortosa et al., 2017*), dendritic arborization (*Takemoto-Kimura et al., 2007*), spine formation (*George et al., 2015*; *Kang et al., 2008*; *Kutzleb et al., 1998*), synapse assembly, synaptic transmission (*El-Husseini et al., 2002*; *Hayashi et al., 2005*; *Keith et al., 2012*; *Lin et al., 2009*; *Sanders et al., 2020*; *Thomas et al., 2012*), and synaptic plasticity (*Brigidi et al., 2014*). Strikingly, there is an apparent interaction between palmitoylation and synaptic transmission since neuronal activity can regulate protein palmitoylation (*Brigidi et al., 2014*; *Hayashi et al., 2005*; *Kang et al., 2008*). Palmitoylation is catalyzed by palmitoyl acyltransferases (PATs), enzymes that harbor the catalytically active Asp-His-His-Cys (DHHC) signature motif (thereby, also referred to as DHHCs). Originally discovered in the yeast *Saccharomyces cerevisiae* (*Lobo et al., 2002*; *Roth et al., 2002*), thus far 23 DHHC enzymes are predicted to exist in humans and mice. PATs exhibit overlapping, yet distinct, specificities for their substrates (*Huang et al., 2004*; *Roth et al., 2006*). Despite systematic and extensive efforts (*Roth et al., 2006*), our knowledge of DHHC enzyme substrates remains very limited, hindering our understanding of the roles that PATs play in neuronal development and function.

Here, we show that both Nrp2 and Nrp1 are palmitoylated in cortical neurons in vitro and in vivo, and that palmitoylation of select Nrp2 cysteine residues is required for correct Nrp2 subcellular localization, cell surface clustering, and for Sema3F/Nrp2-dependent dendritic spine pruning both in vitro and in vivo. Our findings also reveal that the PAT ZDHHC15 is required for proper Nrp2 palmitoylation and Sema3F/Nrp2-dependent dendritic spine pruning in layer V cortical pyramidal neurons, but not for Nrp1 palmitoylation or for Sema3A/Nrp1-dependent basal dendrite elaboration in these same neurons. These results highlight the importance of guidance cue receptor posttranslational lipid modifications to regulate distinct aspects of cortical neuron morphology.

## Results

### Nrp2 and Nrp1 exhibit distinct cell surface distribution patterns and global inhibition of palmitoylation leads to Nrp2 mislocalization

To understand the mechanisms by which Sema3F and Sema3A exert distinct effects on the development of layer V cortical pyramidal neurons, we investigated the localization and function of their obligate co-receptors, Nrp2 and Nrp1, respectively. First, we assessed the localization of Nrp2 and Nrp1 on the surface of COS-7 cells owing to the well-articulated plasma membrane and large circumference of these cells in culture. COS-7 cells transfected with flag-tagged wild-type Nrp2 or Nrp1 expression plasmids were subjected to surface staining with a flag antibody to visualize Nrp localization on the plasma membrane. We observed that Nrp2 is distributed on the COS-7 cell surface in a clustered pattern consisting of numerous discrete puncta (*Figure 1A*), whereas Nrp1 is evenly distributed over the entire plasma membrane with little evidence of clustering (*Figure 1B*). Particle analysis (see Materials and methods) provides a quantitative assessment of protein clustering and reveals a marked difference between the membrane localization of Nrp2 and Nrp1 (*Figure 1C*). This robust Nrp2 clustering is reminiscent of the cell surface clustering displayed by palmitoylated proteins in COS-7 cells (*Webb et al., 2000*).

To explore the potential role of palmitoylation in Nrp2 surface localization, we used a pharmacological approach employing 2-bromopalmitate, which is a specific and irreversible inhibitor of protein palmitoylation (*Jennings et al., 2009*; *Resh, 2006*). We treated COS-7 cells expressing flag-tagged wild-type Nrp2 (Nrp2$^{WT}$) with 2-bromopalmitate, or a control solution, and visualized cell surface Nrp2. Following bath application of 2-bromopalmitate, Nrp2 was markedly redistributed compared to the control, displaying a diffuse localization on the cell surface similar to Nrp1, which is localized diffusely following either control or 2-bromopalmitate treatment (*Figure 1D and E*). 2-Bromopalmitate-induced Nrp2 dispersion is consistent with previous reports showing 2-bromopalmitate-induced diffusion of otherwise clustered palmitoylated proteins in non-neuronal cell lines (*Webb et al., 2000*) and in neurons (*El-Husseini et al., 2002*). These results show that Nrp2 and Nrp1 exhibit distinct cell surface compartmental localization and that protein palmitoylation is required for Nrp2 cell surface clustering in heterologous cells in vitro.

### Nrp2 and Nrp1 are palmitoylated in cortical neurons in vitro and in the mouse brain, exhibiting overlapping palmitoylation patterns

Given the significant Nrp2 cell surface distribution perturbations observed upon 2-bromopalmitate bath application to COS-7 cells, we next examined whether Nrps are palmitoylated. Both Nrp2 and Nrp1 harbor cysteine residues in their transmembrane and membrane-proximal (juxtamembrane) domains (*Figure 2A*, in red). Nrp2 also has a lone cysteine residue, C897 (*Figure 2A*, in red) carboxy-terminal (C-terminal) to the transmembrane domain and two adjacent C-terminal cysteine residues (*Figure 2A*, in blue). These Nrp2 and Nrp1 cysteine residues are all highly phylogenetically conserved among vertebrate species (*Figure 2—figure supplement 1*). The location and conservation of these cysteine residues, coupled with the presence of palmitoyl acceptor amino acid residues predicted using CSS-Palm software (*Ren et al., 2008*), suggested that these cysteine residues are palmitoyl acceptor sites. To explore this further, we used the acyl-biotin exchange (ABE) assay, a well-established biochemical approach for the detection of palmitoylation on thiol groups of cysteines (S-palmitoylation) (*Drisdel and Green, 2004*; *Kang et al., 2008*; *Roth et al., 2006*; *Wan et al., 2007*). In this assay, we used the postsynaptic density protein 95 (PSD-95) as a positive palmitoylation control because it is palmitoylated (*Topinka and Bredt, 1998*), and synapse-associated protein 102 (SAP102) as a negative palmitoylation control because it is not (*Kang et al., 2008*). We observed that both Nrp2 and Nrp1 are palmitoylated in adult mouse forebrain and in embryonic day 14.5 (E14.5) mouse cerebral cortex (*Figure 2B*), as are both Nrps in cortical neurons 28 days in vitro (DIV) derived from E14.5 embryos (*Figure 2C*). The detection of Nrp palmitoylation from early embryonic stages (E14.5) to adulthood raises the possibility that palmitoylation of these co-receptors is required throughout neural development and in the adult to regulate Nrp trafficking and function.

We next sought to identify the Nrp2 and Nrp1 cysteine residues that serve as palmitate acceptors. For this analysis, we generated point mutations that replaced individual Nrp2 and Nrp1 cysteines with a serine residue, which cannot be S-palmitoylated. Cysteine-to-serine (CS) point mutant Nrp

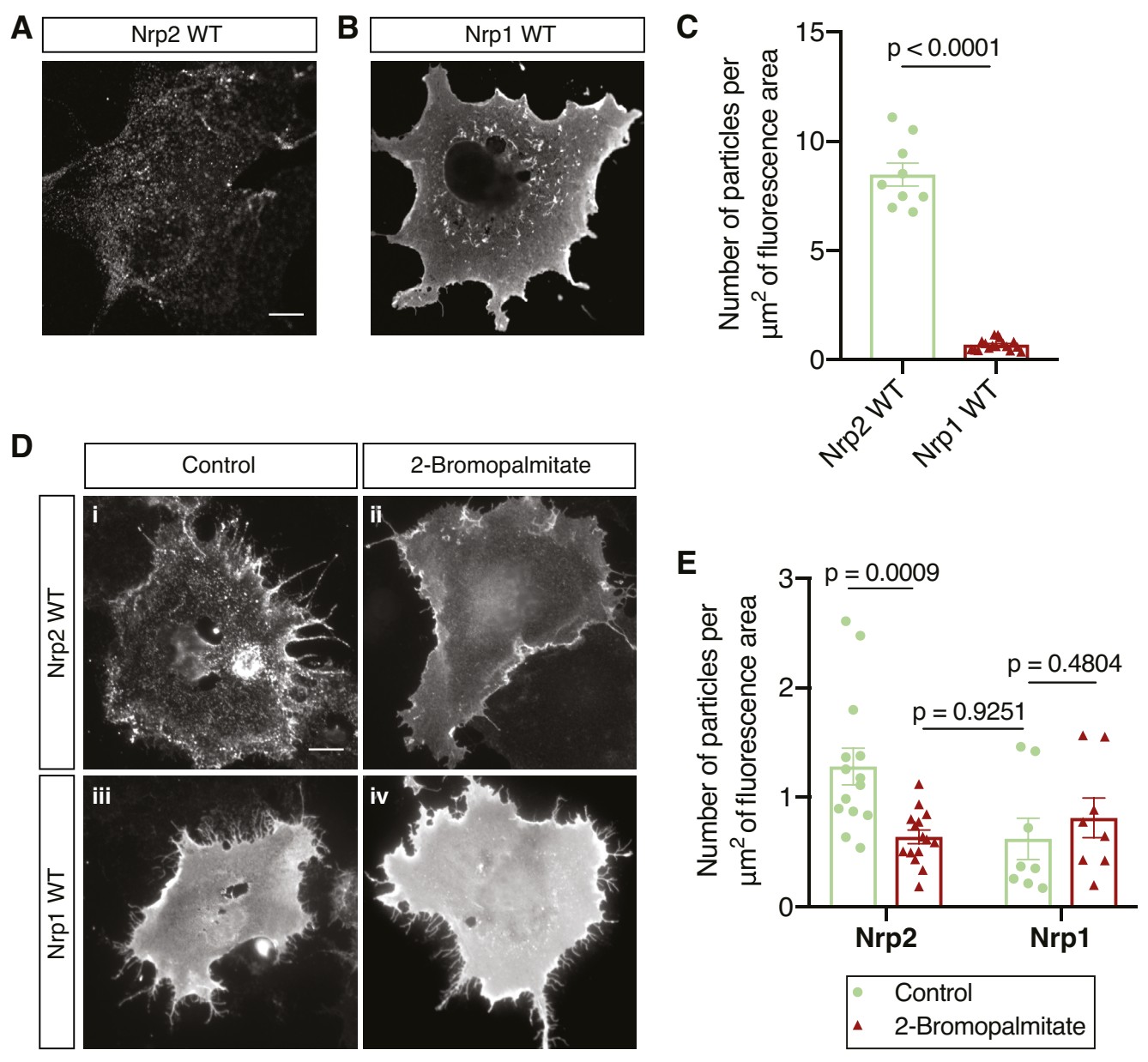

**Figure 1.** Distinct cell surface localization patterns of neuropilin-2 (Nrp2) and Nrp1 are abolished by global inhibition of palmitoylation. (**A–C**) Surface localization of Nrp2 and Nrp1 in COS-7 cells expressing exogenous flag-tagged Nrp2 wild-type (WT) (**A**) or Nrp1 WT (**B**). Cells were subjected to surface staining with a flag antibody to visualize cell surface protein. Representative images (single plane) are shown for each protein. Nrp2 appears clustered, whereas Nrp1 appears diffusely localized on the cell surface. Scale bar, 12 μm. (**C**) Quantification of protein clustering, expressed as the number of particles per μm² of fluorescence area (Clustering analysis, see Materials and methods). Data are plotted in a scatter dot plot with mean ± SEM (SEM, standard error of the mean). Two-tailed t test; Nrp2, n=9; Nrp1, n=17, where n is the number of cells analyzed. (**D, E**) Effects of 2-bromopalmitate on cell surface Nrp localization in heterologous cells. COS-7 cells expressing exogenous flag-tagged WT Nrp2 or Nrp1 were treated overnight with medium containing either 10 μM 2-bromopalmitate (ii, iv) or the same concentration of solvent (i, iii). Cells were subjected to surface staining with a flag antibody. Representative images (single plane) are shown for each protein. Upon control treatment, Nrp2 appears highly clustered (**i**), while Nrp1 has an even diffuse distribution on the plasma membrane (**iii**). Upon treatment with 2-bromopalmitate, Nrp2 assumes diffuse distribution (**ii**), similar to Nrp1 (**iv**). Scale bar, 20 μm. (**E**) Quantification of protein clustering shown in (**D**), as mentioned above. Data are plotted in a scatter dot plot with mean ± SEM. Two-tailed t test; Nrp2 control, n=14; Nrp2 2-bromopalmitate, n=15; Nrp1 control, n=8; Nrp1 2-bromopalmitate, n=8, where n is the number of cells analyzed.

The online version of this article includes the following source data for figure 1:

**Source data 1.** Raw data for *Figure 1C*.

**Source data 2.** Raw data for *Figure 1E*.

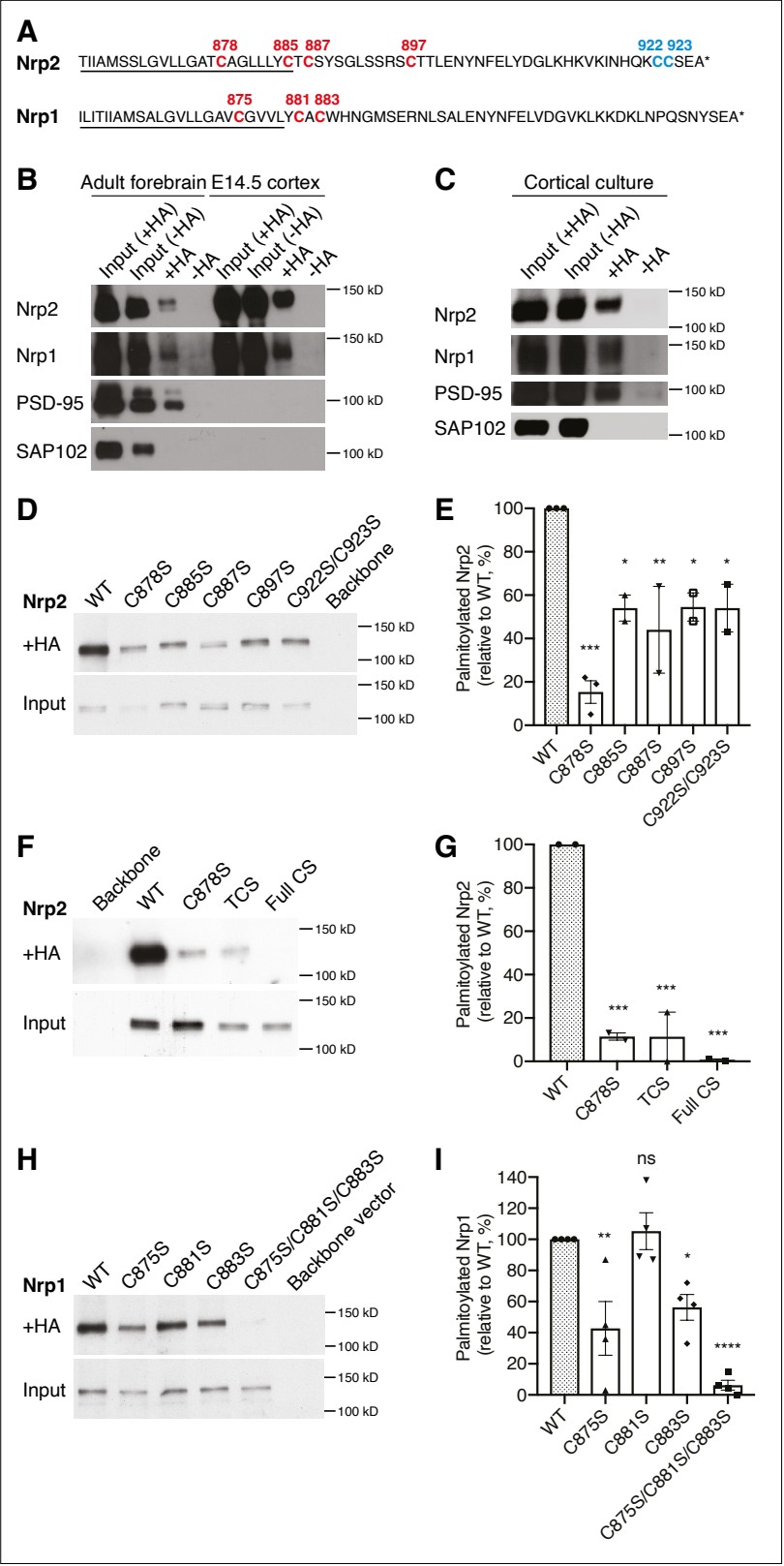

**Figure 2.** Neuropilins are palmitoylated in deep layer primary cortical neurons and in the mouse brain. (**A**) Amino acid sequence of transmembrane (underlined) and cytoplasmic domains of neuropilin-2 (Nrp2) and Nrp1, as predicted by Ensembl genome database. Both Nrp2 and Nrp1 harbor cysteine residues in their transmembrane and juxtamembrane domains (depicted in red), whereas Nrp2 also possesses a C-terminal di-cysteine motif

*Figure 2 continued on next page*

*Figure 2 continued*

(depicted in blue). Cysteine numeration accords with their position in the amino acid sequence of *Mus musculus* Nrp2 (Nrp2-202, isoform A17, ENSMUST00000063594.13) for Nrp2 and *M. musculus* Nrp1 (Nrp1-201, ENSMUST00000026917.10) for Nrp1. (**B, C**) Acyl-biotin exchange (ABE) performed on adult mouse forebrain or embryonic day 14.5 (E14.5) cerebral cortex (**B**) and on E14.5 days in vitro (DIV)28 cortical neuron cultures (**C**). Western blots show endogenous proteins that are detected with protein-specific antibodies. Hydroxylamine-treated (+HA) samples show palmitoylated protein, while -HA samples serve as a negative control. Separate inputs were taken from the samples that were then processed with +HA or -HA buffer; therefore 'Input (+HA)' represents the input of +HA sample and 'Input (-HA)' represents the input of -HA sample. Both Nrp2 and Nrp1 are palmitoylated in the adult mouse forebrain, in E14.5 cerebral cortex and in cortical neuron cultures, similar to postsynaptic density protein 95 (PSD-95) that serves as a positive palmitoylation control, as shown by the presence of signal in the +HA sample and the absence of signal in the -HA control sample. Synapse-associated protein 102 (SAP102), known not to be palmitoylated, serves as a negative control for palmitoylation and it is not palmitoylated. At E14.5, no PSD-95 or SAP102 are detected in the mouse cerebral cortex, apparently due to the very low expression of these synaptic markers at this stage of development. (**D, E**) ABE on neuroblastoma-2a (Neuro-2a) cells expressing flag-tagged Nrp2 wild-type (WT) or CS point mutants. (**D**) Nrp2 immunoblots show palmitoylated (+HA) and input samples. -HA samples (not shown) were also included in the experiment. (**E**) Quantification of palmitoylated protein levels (fraction of the protein that is palmitoylated) calculated as the ratio of +HA to the respective input and plotted in a scatter dot plot with mean ± SEM (n=2–3 experiments). Palmitolyated WT Nrp2 is set at 100% and CS mutants are expressed as a percentage of WT. The cysteine residue C878 stands out as a major Nrp2 palmitate acceptor site. One-way analysis of variance (ANOVA) followed by Dunnett's test for multiple comparisons (Nrp2 CS mutants are compared to Nrp2 WT [set at 100%]: C878S, p=0.0002; C885S, p=0.0189; C887S, p=0.0062; C897S, p=0.0200; C922S/C923S, p=0.0189). (**F, G**) ABE on Neuro-2a cells expressing flag-tagged Nrp2 WT or various CS mutants. (**F**) Nrp2 immunoblots show palmitoylated (+HA) and input samples. -HA samples (not shown) were also included in the experiment. (**G**) Quantification of palmitoylated protein levels (as explained above), plotted in scatter dot plots with mean ± SEM (n=2 experiments). The Nrp2 transmembrane and membrane-proximal cysteines are major palmitoylation sites. One-way ANOVA followed by Dunnett's test for multiple comparisons (Nrp2 CS mutants are compared to Nrp2 WT [set at 100%]: C878S, p=0.0009; TCS, p=0.0009; Full CS, p=0.0006). (**H, I**) ABE on Neuro-2a cells expressing flag-tagged Nrp1 WT or CS point mutants. (**H**) Nrp1 immunoblots show palmitoylated (+HA) and input samples. -HA samples (not shown) were also included in the experiment. (**I**) Quantification of palmitoylated protein levels (as explained above), plotted in scatter dot plot including mean ± SEM (n=4 experiments). Nrp1 is palmitoylated mostly on cysteines C875 and C883. One-way ANOVA followed by Dunnett's test for multiple comparisons (Nrp1 CS mutants are compared to Nrp1 WT (set at 100%): C875S, p=0.0042; C881S, p=0.9877; C883S, p=0.0272; C875S/C881S/C883S, p<0.0001; ns, not significant).

The online version of this article includes the following source data and figure supplement(s) for figure 2:

**Source data 1.** Raw, unedited blot from *Figure 2B*.

**Source data 2.** Raw, unedited blot from *Figure 2B*.

**Source data 3.** Raw, unedited blot from *Figure 2B*.

**Source data 4.** Raw, labeled blot from *Figure 2B*.

**Source data 5.** Raw, labeled blot from *Figure 2B*.

**Source data 6.** Raw, labeled blot from *Figure 2B*.

**Source data 7.** Raw, unedited blot from *Figure 2C*.

**Source data 8.** Raw, unedited blot from *Figure 2C*.

**Source data 9.** Raw, unedited blot from *Figure 2C*.

**Source data 10.** Raw, unedited blot from *Figure 2C*.

**Source data 11.** Raw, labeled blot from *Figure 2C*.

**Source data 12.** Raw, labeled blot from *Figure 2C*.

**Source data 13.** Raw, labeled blot from *Figure 2C*.

**Source data 14.** Raw, labeled blot from *Figure 2C*.

**Source data 15.** Raw, unedited blot from *Figure 2D*.

**Source data 16.** Raw, labeled blot from *Figure 2D*.

**Source data 17.** Raw data for *Figure 2E*.

**Source data 18.** Raw, unedited blot from *Figure 2F*.

**Source data 19.** Raw, unedited blot from *Figure 2F*.

**Source data 20.** Raw, labeled blot from *Figure 2F*.

*Figure 2 continued*

**Source data 21.** Raw, labeled blot from *Figure 2F*.

**Source data 22.** Raw data for *Figure 2G*.

**Source data 23.** Raw, unedited blot from *Figure 2H*.

**Source data 24.** Raw, labeled blot from *Figure 2H*.

**Source data 25.** Raw data for *Figure 2I*.

**Figure supplement 1.** Conserved cysteine residues lie in the transmembrane and cytoplasmic domains of neuropilins.

expression plasmids were transfected into neuroblastoma-2a (Neuro-2a) cells on which the ABE assay was performed. This analysis revealed that Nrp2 is palmitoylated on all transmembrane and membrane-proximal cysteines, albeit to variable extents, as well as on the C-terminal di-cysteine motif (*Figure 2D and E*). The triple CS (TCS) Nrp2 mutant C878S/C885S/C887S (Nrp2$^{TCS}$: substituting serine for the three transmembrane/juxtamembrane cysteines) and the Full CS Nrp2 mutant C878S/C885S/C887S/C897S/C922S/C923S (Nrp2$^{Full\ CS}$: substituting all transmembrane and cytoplasmic cysteines) display very little to no palmitoylation, respectively (*Figure 2F and G*). A similar analysis for Nrp1 revealed that Nrp1 is palmitoylated on cysteines C875 and C883, but apparently not on C881 (*Figure 2H and I*). These results show that Nrp2 and Nrp1 display similar palmitoylation patterns in their transmembrane and membrane-proximal segments, and that Nrp2 also harbors C-terminal palmitoylation sites that are not present in Nrp1.

## Select palmitoyl acceptor cysteines regulate the localization and trafficking of Nrp2 across subcellular compartments

Palmitoylation enables proteins to be anchored onto specialized membrane domains, to interact with other proteins, and to shuttle between the plasma membrane and intracellular organelles (*Salaun et al., 2010*). The compartmentalized distribution of Nrp2 on the cell surface of COS-7 cells in culture, which is palmitoylation-dependent (*Figure 1D and E*), prompted us to investigate the role of palmitoylation in Nrp2 localization, trafficking, and function in cortical neurons.

First, we addressed the role of palmitoyl acceptor Nrp2 cysteines in Nrp2 surface localization in cell lines. We expressed wild-type or various CS mutant flag-tagged Nrp2 proteins in COS-7 cells and assessed their plasma membrane distribution with surface staining. This assay revealed that localization of Nrp2$^{C922S/C923S}$, which lacks the C-terminal di-cysteine motif, is for the most part punctate with little difference from Nrp2$^{WT}$, whereas Nrp2$^{TCS}$ and Nrp2$^{Full\ CS}$ proteins are profoundly mislocalized (*Figure 3—figure supplement 1A–1C*), exhibiting a non-patterned distribution over the entire plasma membrane that resembles Nrp1. For our localization analysis experiments, we also generated pHluorin-tagged Nrp2 wild-type and CS mutants. The performance of the same localization assay with pHluorin-tagged Nrp2 wild-type and CS mutants yielded similar results (*Figure 3A and B*). The diffuse distribution of palmitoylation-deficient Nrp2 CS mutants on the cell surface is reminiscent of Nrp2$^{WT}$ protein distribution following treatment with 2-bromopalmitate (*Figure 1D*). Further, these results are in line with the palmitoylation patterns we observed for these same Nrp2 CS mutants (*Figure 2D–G*).

Second, we examined Nrp2 cell surface localization in cortical neurons in culture. We transfected cortical neurons with plasmids expressing pHluorin-tagged wild-type Nrp2 or various CS mutants and carried out live imaging. The pH-sensitive GFP variant pHluorin fluoresces robustly at neutral pH, and so the tagged protein is readily visualized when localized at the cell surface (see Materials and methods). These live imaging assessments of surface protein localization showed that Nrp2$^{WT}$ (*Figure 3—figure supplement 2A*; *Video 1*) and Nrp2$^{C922S/C923S}$ (*Figure 3—figure supplement 2C*) exhibit a prominent punctate localization along the plasma membrane, consistent with their distribution in COS-7 cells. In contrast, Nrp2$^{TCS}$ apparently forms larger protein clusters (*Figure 3—figure supplement 2B*), and Nrp2$^{Full\ CS}$ is profoundly mislocalized displaying a diffuse distribution over all membranes (*Figure 3—figure supplement 2D*). These pHluorin-tagged Nrp2 plasmids were also tested with live imaging in Neuro-2a cells, with similar results (data not shown).

Given the surface mislocalization of palmitoylation-deficient Nrp2, we next addressed the early intracellular trafficking defects that lead to aberrant Nrp2 delivery to the cell surface. Since the Golgi apparatus is the main intracellular compartment where palmitoylation enzymes are localized

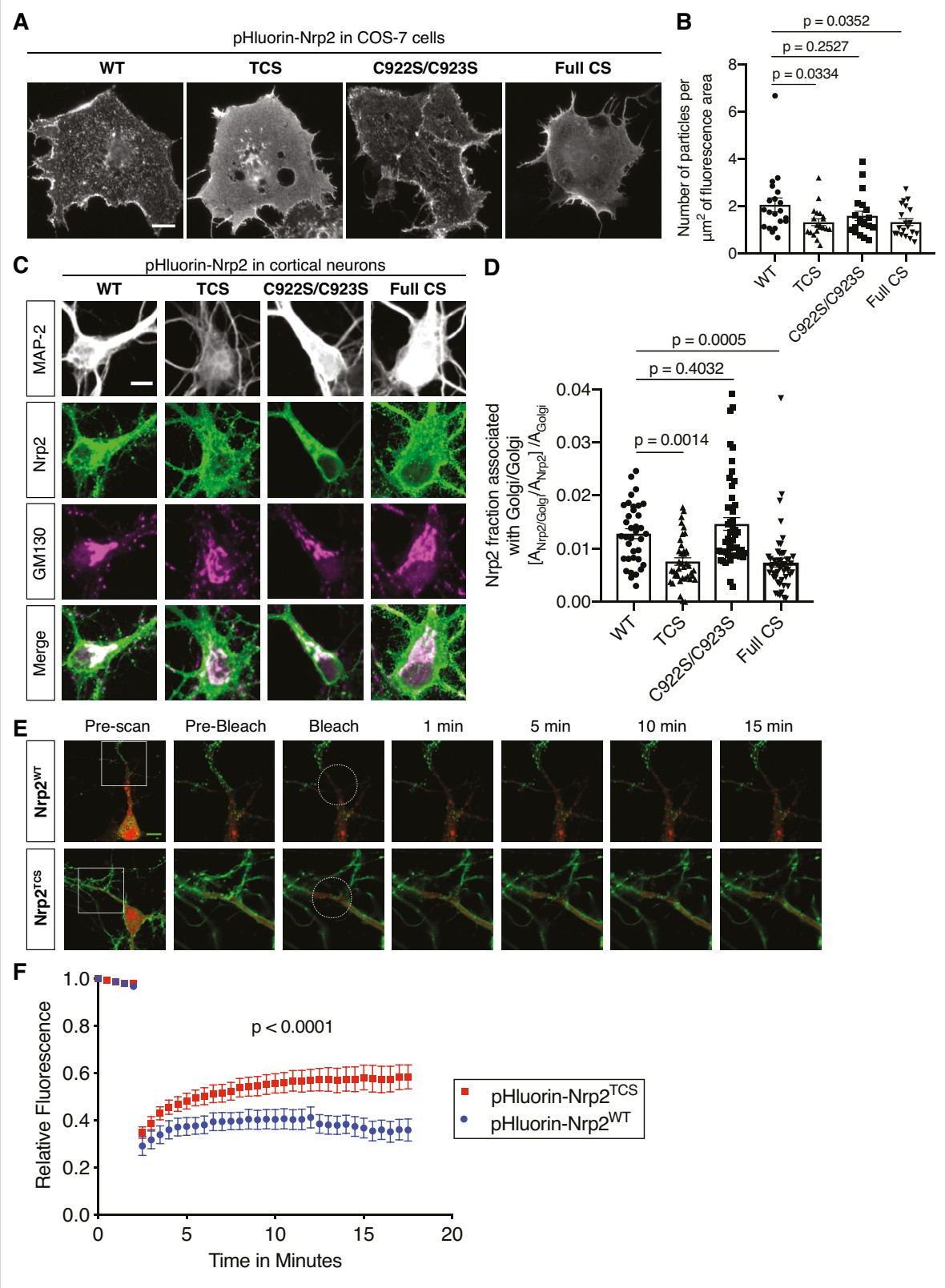

**Figure 3.** Differential requirements for distinct neuropilin-2 (Nrp2) palmitoyl acceptor cysteines in regulating subcellular Nrp2 localization and trafficking. (**A, B**) Effects of Nrp2 cysteines on Nrp2 protein cell surface localization in heterologous cells. (**A**) Panels show COS-7 cells expressing pHluorin-tagged Nrp2 wild-type (WT) or various cysteine-to-serine (CS) mutants and subjected to Nrp2 surface staining with a GFP antibody. Scale bar, 15 µm. (**B**) Quantification of protein clustering with particle analysis (as mentioned earlier), expressed as number of particles per µm² of fluorescence area and

*Figure 3 continued on next page*

*Figure 3 continued*

plotted in scatter dot plot including mean ± SEM. WT and C922S/C923S Nrp2 proteins are distributed in the form of smaller particles (puncta), whereas triple CS (TCS) and Full CS Nrp2 proteins localize on the surface as large protein clusters. One-way analysis of variance (ANOVA) followed by Dunnett's test for multiple comparisons; n=20 cells for each plasmid. (**C, D**) Effects of Nrp2 cysteines on Nrp2 localization at the Golgi apparatus in cortical neuron cultures. Assessment of colocalization of pHluorin-tagged Nrp2 WT or CS mutants with the Golgi apparatus marker Golgi matrix protein 130 (GM130) in embryonic day 14.5 (E14.5) days in vitro (DIV)17 *Nrp2⁻/⁻* primary cortical neurons. (**C**) Representative images of neurons (single plane) expressing different Nrp2 proteins. Nrp2 here is visualized via EGFP immunofluorescence, and the signal represents total Nrp2 (surface and intracellular). In merged panels, the regions where Nrp2 and GM130 puncta of similar intensity colocalize appear white. Nrp2 WT and C922S/C923S exhibit very strong association with Golgi cisternae. By contrast, Nrp2 TCS or Full CS colocalize with the Golgi to a significantly lesser extent. Of note, following Nrp2 staining, neurons expressing Full CS Nrp2 appear 'hairy'; this is a common phenotype of this Nrp2 CS mutant, indicative of its diffuse distribution in all membranes including filopodia. Scale bar, 7 μm. (**D**) Quantification of the colocalization between Nrp2 and GM130, expressed as the fraction of Nrp2 associated with Golgi [$A_{Nrp2/Golgi}/A_{Nrp2}$] normalized to the quantity of Golgi (Golgi Area, $A_{Golgi}$) present in each neuron ([$A_{Nrp2/Golgi}/A_{Nrp2}$]/$A_{Golgi}$) (see Materials and methods). Columns show pooled data from independent cultures plotted in scatter dot plot including mean ± SEM. One-way ANOVA (p<0.0001) followed by Dunnett's test for multiple comparisons; WT, 36 neurons; TCS, 40 neurons; C922S/C923S, 50 neurons; Full CS, 50 neurons. (**E, F**) Fluorescence recovery after photobleaching (FRAP) analysis, on E14.5 DIV10 WT cortical neurons expressing pHluorin-tagged *Nrp2 WT* or *TCS* plasmids. (**E**) Time-lapse image sequences are shown for each Nrp2 protein. Pre-bleach and post-bleach panels depict the areas surrounded by the white squares in pre-scan panels. White dashed circles delineate the region of interest (ROI) selected for photobleaching. Note the higher diffusibility of Nrp2^TCS compared to Nrp2^WT and the difference in their surface distributions that appear diffuse and clustered, respectively. Scale bar, 10 μm. (**F**) Quantitative analysis of fluorescence recovery kinetics after photobleaching. Pooled data are plotted as mean ± SEM. Extra sum-of-squares F test; WT: 7 neurons, Rmax = 0.4463, T1/2=1.242e-016; TCS: 8 neurons, Rmax = 0.5824, T1/2=1.348e-016, where Rmax is maximum recovery.

The online version of this article includes the following source data and figure supplement(s) for figure 3:

**Source data 1.** Raw data for *Figure 3B*.

**Source data 2.** Raw data for *Figure 3D*.

**Source data 3.** Raw data for *Figure 3F*.

**Figure supplement 1.** Distinct requirements for neuropilin-2 (Nrp2) palmitoyl acceptor cysteines in Nrp2 cell surface distribution in COS-7 cells.

**Figure supplement 1—source data 1.** Raw data for *Figure 3—figure supplement 1B and C*.

**Figure supplement 2.** Severe defects in the cell surface localization of palmitoylation-deficient neuropilin-2 (Nrp2) in primary cortical neurons.

**Figure supplement 3.** Neuropilin-2 (Nrp2) is enriched in the Golgi apparatus in neural tissue.

**Figure supplement 3—source data 1.** Raw, unedited blot from *Figure 3—figure supplement 3B*.

**Figure supplement 3—source data 2.** Raw, unedited blot from *Figure 3—figure supplement 3B*.

**Figure supplement 3—source data 3.** Raw, labeled blot from *Figure 3—figure supplement 3B*.

**Figure supplement 3—source data 4.** Raw, labeled blot from *Figure 3—figure supplement 3B*.

**Figure supplement 4.** Neuropilin-2 (Nrp2) palmitoyl acceptor cysteines are not required for Nrp2/plexinA3 (PlexA3) association but are required for proper Nrp2 homo-oligomerization.

**Figure supplement 4—source data 1.** Raw, unedited blot from *Figure 3—figure supplement 4A*.

**Figure supplement 4—source data 2.** Raw, unedited blot from *Figure 3—figure supplement 4A*.

**Figure supplement 4—source data 3.** Raw, unedited blot from *Figure 3—figure supplement 4A*.

**Figure supplement 4—source data 4.** Raw, unedited blot from *Figure 3—figure supplement 4A*.

**Figure supplement 4—source data 5.** Raw, labeled blot from *Figure 3—figure supplement 4A*.

**Figure supplement 4—source data 6.** Raw, labeled blot from *Figure 3—figure supplement 4A*.

**Figure supplement 4—source data 7.** Raw, labeled blot from *Figure 3—figure supplement 4A*.

**Figure supplement 4—source data 8.** Raw, labeled blot from *Figure 3—figure supplement 4A*.

**Figure supplement 4—source data 9.** Raw data for *Figure 3—figure supplement 4B*.

**Figure supplement 4—source data 10.** Raw, unedited blot from *Figure 3—figure supplement 4D*.

**Figure supplement 4—source data 11.** Raw, unedited blot from *Figure 3—figure supplement 4D*.

**Figure supplement 4—source data 12.** Raw, unedited blot from *Figure 3—figure supplement 4D*.

**Figure supplement 4—source data 13.** Raw, labeled blot from *Figure 3—figure supplement 4D*.

**Figure supplement 4—source data 14.** Raw, labeled blot from *Figure 3—figure supplement 4D*.

**Figure supplement 4—source data 15.** Raw, labeled blot from *Figure 3—figure supplement 4D*.

**Figure supplement 4—source data 16.** Raw data for *Figure 3—figure supplement 4E*.

**Figure supplement 4—source data 17.** Raw, unedited blot from *Figure 3—figure supplement 4F*.

**Figure supplement 4—source data 18.** Raw, unedited blot from *Figure 3—figure supplement 4F*.

**Figure supplement 4—source data 19.** Raw, unedited blot from *Figure 3—figure supplement 4F*.

*Figure 3 continued*

**Figure supplement 4—source data 20.** Raw, labeled blot from *Figure 3—figure supplement 4F*.

**Figure supplement 4—source data 21.** Raw, labeled blot from *Figure 3—figure supplement 4F*.

**Figure supplement 4—source data 22.** Raw, labeled blot from *Figure 3—figure supplement 4F*.

**Figure supplement 4—source data 23.** Raw data for *Figure 3—figure supplement 4G*.

and function (*Ohno et al., 2006*; *Rocks et al., 2010*), we asked whether Nrp2 localizes to Golgi membranes. We stained primary cortical neurons in culture with antibodies directed against Nrp2 and the *cis*-Golgi marker Golgi matrix protein 130 (GM130), which showed that endogenous Nrp2 is robustly associated with somatic Golgi and dendritic Golgi cisternae known as Golgi outposts (*Figure 3—figure supplement 3A*). Further, Golgi isolation experiments from mouse brain revealed Nrp2 enrichment in the GM130-positive Golgi fraction (*Figure 3—figure supplement 3B*).

Next, we examined whether palmitoylated Nrp2 cysteine residues are required for the association between Nrp2 and the Golgi apparatus by performing immunofluorescence on $Nrp2^{-/-}$ cortical neurons transfected with either wild-type Nrp2 or various Nrp2 CS mutants. These experiments show that $Nrp2^{WT}$ and $Nrp2^{C922S/C923S}$ proteins display robust association with Golgi membranes, whereas the $Nrp2^{TCS}$ and $Nrp2^{Full CS}$ mutant proteins are significantly deficient in their association with the Golgi apparatus (*Figure 3C, D*). The aberrant association of TCS and Full CS Nrp2 proteins with the Golgi apparatus could have major effects on their trafficking and plasma membrane insertion.

To investigate the role of palmitoylation in the spatial and temporal dynamics of Nrp2 trafficking, we focused on the role of transmembrane/juxtamembrane cysteines, which have the greatest effect on Nrp2 surface localization and Golgi association, performing fluorescence recovery after photobleaching (FRAP) in cortical neurons expressing exogenous pHluorin-tagged $Nrp2^{WT}$ or $Nrp2^{TCS}$. The fluorescent signal was bleached over a selected region of interest (ROI) and after photobleaching neurons were imaged for at least 15 min to visualize the recovery of fluorescence in the bleached area, which represents protein molecules either derived from nearby plasma membrane compartments or newly inserted in the cell membrane. We observed that $Nrp2^{TCS}$ protein displays a markedly different trafficking pattern with a higher mobile fraction compared to $Nrp2^{WT}$ (*Figure 3E and F*; *Videos 2 and 3*). This observation is in accordance with other studies reporting higher diffusibility and an altered intracellular localization of depalmitoylated proteins (*Miura et al., 2006*; *Rocks et al., 2010*; *Rocks et al., 2005*).

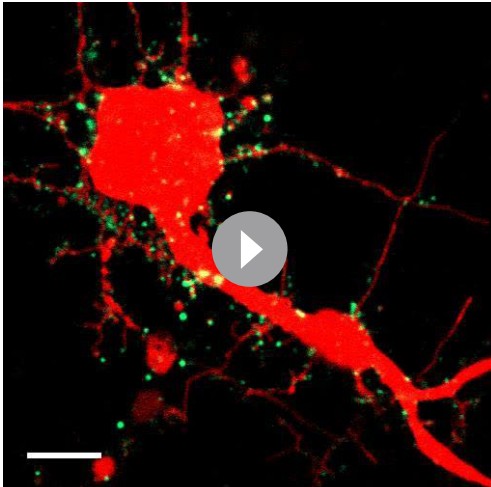

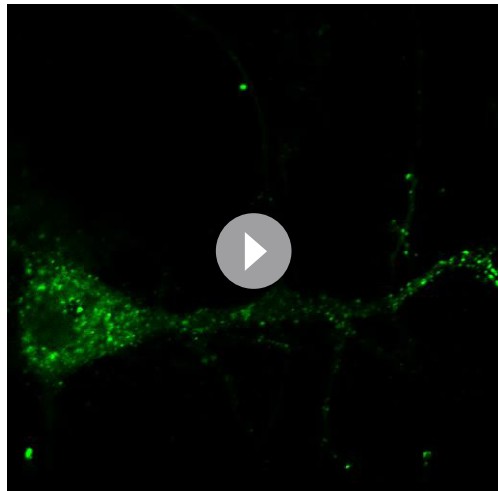

**Video 1.** Neuropilin-2 (Nrp2) localizes on the plasma membrane of primary cortical neurons in numerous discrete puncta. Time-lapse imaging of a neuron expressing pHluorin-tagged Nrp2 WT. Scale bar, 8 µm.
https://elifesciences.org/articles/83217/figures#video1

**Video 2.** Trafficking dynamics of pHluorin-tagged Nrp2 WT in primary cortical neurons visualized with fluorescence recovery after photobleaching (FRAP) (from *Figure 3E*).
https://elifesciences.org/articles/83217/figures#video2

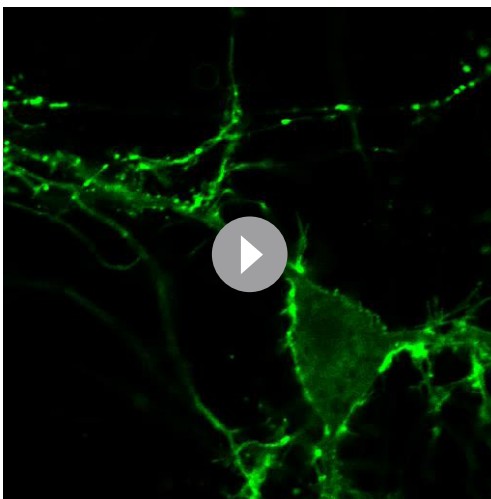

**Video 3.** Trafficking dynamics of pHluorin-tagged Nrp2 TCS in primary cortical neurons visualized wtih fluorescence recovery after photobleaching (FRAP) (From *Figure 3E*).

https://elifesciences.org/articles/83217/figures#video3

Taken together, these experiments support the functional importance of transmembrane/juxtambembrane palmitoyl acceptor Nrp2 cysteines in regulating the localization and trafficking of Nrp2 protein across subcellular compartments and the plasma membrane.

## Nrp2 palmitoyl acceptor cysteines are dispensable for the association between Nrp2 and PlexA3 but are required for Nrp2 homo-oligomerization

Sema3F binds Nrp2, which forms homo-oligomers and interacts with plexinA3 (PlexA3) to form the Sema3F holoreceptor complex (*Giger et al., 1998*; *Janssen et al., 2012*). Since cysteine residues and their posttranslational modification can influence protein structure, we asked whether Nrp2 cysteines control the ability of Nrp2 to associate with PlexA3 or affect the ability of Nrp2 to homo-oligomerize.

First, we performed coimmunoprecipitation experiments using myc-tagged PlexA3 and either Nrp2$^{WT}$ or Nrp2$^{Full\ CS}$. These experiments showed that PlexA3 binds Nrp2$^{Full\ CS}$ to a similar extent as it does to Nrp2$^{WT}$ (*Figure 3—figure supplement 4A and B*). Further investigation into the association between palmitoylation-deficient Nrp2 and PlexA3 using immunofluorescence in primary cortical neurons in culture showed that exogenous Nrp2$^{Full\ CS}$ exhibits strong co-localization with endogenous PlexA3, similar to exogenously expressed Nrp2$^{WT}$ (*Figure 3—figure supplement 4C*). Thus, the membrane-spanning and cytoplasmic Nrp2 cysteines are dispensable for the interaction between Nrp2 and PlexA3.

Next, we addressed whether the transmembrane and cytoplasmic Nrp2 cysteines are required for Nrp2 homo-oligomerization. We performed coimmunoprecipitation experiments using flag-tagged Nrp2$^{WT}$ and pHluorin-tagged Nrp2$^{WT}$ or Nrp2$^{Full\ CS}$. These experiments revealed that pHluorin-Nrp2$^{Full\ CS}$ associates with flag-Nrp2$^{WT}$ to a significantly lesser extent than pHuorin-Nrp2$^{WT}$ does (*Figure 3—figure supplement 4D and E*). Moreover, coimmunoprecipitation experiments between flag-Nrp2$^{WT}$ and pHluorin-Nrp2$^{WT}$ or Nrp2$^{TCS}$ demonstrated that pHluorin-Nrp2$^{TCS}$ associates with flag-Nrp2$^{WT}$ to a lesser extent than pHluorin-Nrp2$^{WT}$ does (*Figure 3—figure supplement 4F and G*), although this difference is not as robust as it is with Nrp2$^{Full\ CS}$. These findings show that the Nrp2 cysteine residues that we have shown to be palmitoylated and required for Nrp2 localization are also necessary for Nrp2 homo-oligomerization.

## Transmembrane/juxtamembrane Nrp2 cysteines are required for Sema3F/Nrp2-dependent dendritic spine pruning in deep layer cortical neurons in vitro and in vivo

Sema3F/Nrp2 signaling regulates pruning of supernumerary dendritic spines on apical dendrites of deep layer cortical pyramidal neurons (*Demyanenko et al., 2014*; *Duncan et al., 2021*; *Tran et al., 2009*). To investigate a potential role for palmitoyl acceptor Nrp2 cysteines in Sema3F/Nrp2-dependent spine constraint, we employed an in vitro assay aimed at rescuing the inability of Sema3F to decrease dendritic spine density in Nrp2 knockout (*Nrp2$^{-/-}$*) cortical neurons by introducing flag-tagged Nrp2$^{WT}$ or Nrp2$^{CS}$ mutant expression plasmids subcloned in a *pCIG2-ires-EGFP* backbone vector. *Nrp2$^{-/-}$* primary cortical neurons were transfected with backbone vector as a control or a Nrp2 expression plasmid. At DIV21, when dendritic spines are well developed, neurons were treated with 5 nM Sema3F-AP or 5 nM AP (control) for 6 hr, followed by EGFP immunofluorescence to visualize neuronal morphology. The rescue ability of the various Nrp2 plasmids was assessed using two criteria: (1) a comparison of dendritic spine density between AP-treated *Nrp2$^{-/-}$* neurons expressing backbone vector (control) and AP-treated neurons expressing each *Nrp-2* plasmid; and (2) a comparison of

dendritic spine density between Sema3F-AP-treated and control AP-treated $Nrp2^{-/-}$ cortical neurons expressing a specific (the same) Nrp2 plasmid. If a plasmid is able to rescue the $Nrp2$ knockout spine phenotype, it should: (1) constrain spine number compared to the backbone vector (control) in the absence of exogenous Sema3F treatment and/or (2) cause a reduction in spine number upon exogenous Sema3F-AP treatment compared to AP treatment leading to dendritic spine pruning that is similar to that imparted by $Nrp2^{WT}$ protein.

These experiments show that $Nrp2^{WT}$ and $Nrp2^{C922S/C923S}$ can rescue the $Nrp2$ mutant dendritic spine phenotype (*Figure 4A–C*). This is in accordance with our finding that the C-terminal Nrp2 cysteines regulate Nrp2 subcellular localization to a rather minor extent (*Figure 3A–D*, *Figure 3—figure supplement 1*, *Figure 3—figure supplement 2*). In contrast, $Nrp2^{C878S}$ and $Nrp2^{C887S}$ are not capable of constraining spine number, whereas $Nrp2^{C885S}$ and $Nrp2^{C897S}$ can partially rescue the $Nrp2$ knockout dendritic spine phenotype (*Figure 4A–C*). The relative requirements for Nrp2 palmitoyl acceptor cysteines in Sema3F/Nrp2-dependent dendritic spine pruning are summarized in *Figure 4D*. Likewise, $Nrp2$ mutant rescue experiments in cortical neuron cultures with pHluorin-tagged $Nrp2^{WT}$ or $Nrp2^{CS}$ mutant expression plasmids including the $Nrp2^{TCS}$, $Nrp2^{C922S/C923S}$, and $Nrp2^{Full\ CS}$ mutants, reveal a requirement for transmembrane/juxtamembrane cysteines, but not for the two C-terminal cysteines, in Sema3F/Nrp2-dependent dendritic spine pruning (*Figure 4—figure supplement 1A and B*).

To investigate whether Sema3F effects on cortical neuron dendritic spines are associated with Nrp2 palmitoylation, we asked whether Sema3F bath application impacts Nrp2 palmitoylation. We treated wild-type cortical neurons with 5 nM Sema3F-AP, Sema3A-AP, or AP (control) and assessed palmitoylation of endogenous Nrp2 with the ABE assay. Interestingly, Sema3F application enhances Nrp2 palmitoylation in deep layer primary cortical neurons in culture (*Figure 4E and F*). This shows that Sema3F can upregulate Nrp2 palmitoylation, presumably at nearby subcellular domains and within dendritic spines, and it further suggests that Sema3F effects on cortical neurons are mediated by Nrp2 palmitoylation.

Next, we addressed whether Nrp2 cysteines C878, C885, and C887, which are required for rescuing the $Nrp2$ mutant dendritic spine phenotype in cortical neurons in vitro, are also required for Sema3F/Nrp2-dependent dendritic spine pruning in vivo. We examined the ability of $Nrp2^{TCS}$ to constrain excess dendritic spines resulting from $Nrp2$ deletion in the mouse cortex using an in utero electroporation (IUE) rescue paradigm. To overcome breeding challenges, including reduced viability encountered in the $Nrp2^{-/-}$ mouse line, we crossed homozygous $Nrp2$ mutant mice with homozygous $Nrp2$ floxed ($Nrp2^{F/F}$) mice and targeted layer V cortical pyramidal neurons using IUE in $Nrp2^{F/-}$ embryos of pregnant females at E13.5. To assess the feasibility of this approach, we first carried out two control IUE experiments: one with an EGFP-expressing plasmid to visualize neuronal morphology (control) and a second using a combination of three plasmids: an *EGFP*-expressing plasmid, a *Cre*-expressing plasmid to excise the *floxed Nrp2* allele, and a *LSL-tdTomato*-expressing plasmid that serves as a reporter of Cre activity. In neurons electroporated with all three plasmids, we expected Cre to excise the *floxed Nrp2* allele rendering neurons $Nrp2$ knockout ($Nrp2^{-/-}$) and also to turn on *tdTomato* expression (*Figure 5A*). These experiments revealed increased spine density in neurons lacking Nrp2 compared to control (*Figure 5B and C*), confirming that this approach is a useful assay for rescue analysis of the $Nrp2^{-/-}$-associated dendritic spine phenotype. Next, to assess the rescue ability Nrp2$^{WT}$ and Nrp2$^{TCS}$ proteins in vivo, we used two additional IUE conditions: one with the plasmid combination [*Cre+LSL-tdTomato+flag-Nrp2$^{WT}$-ires-EGFP*] and one with the plasmid combination [*Cre+LSL-tdTomato+flag-Nrp2$^{TCS}$-ires-EGFP*]. Neurons electroporated with the combination expressing Nrp2$^{WT}$ showed significantly fewer spines compared to $Nrp2^{-/-}$ neurons that do not express Nrp2 (electroporated with [*Cre+LSL-tdTomato+EGFP*]). However, neurons electroporated with the combination of plasmids that includes $Nrp2^{TCS}$ exhibited a spine density similar to that of $Nrp2^{-/-}$ neurons (*Figure 5B and C*).

Taken together, both in vitro and in vivo rescue experiments of the $Nrp2^{-/-}$ dendritic spine density phenotype reveal an essential role for the transmembrane and membrane-proximal Nrp2 cysteines in mediating Sema3F/Nrp2-dependent spine pruning in layer V cortical pyramidal neurons.

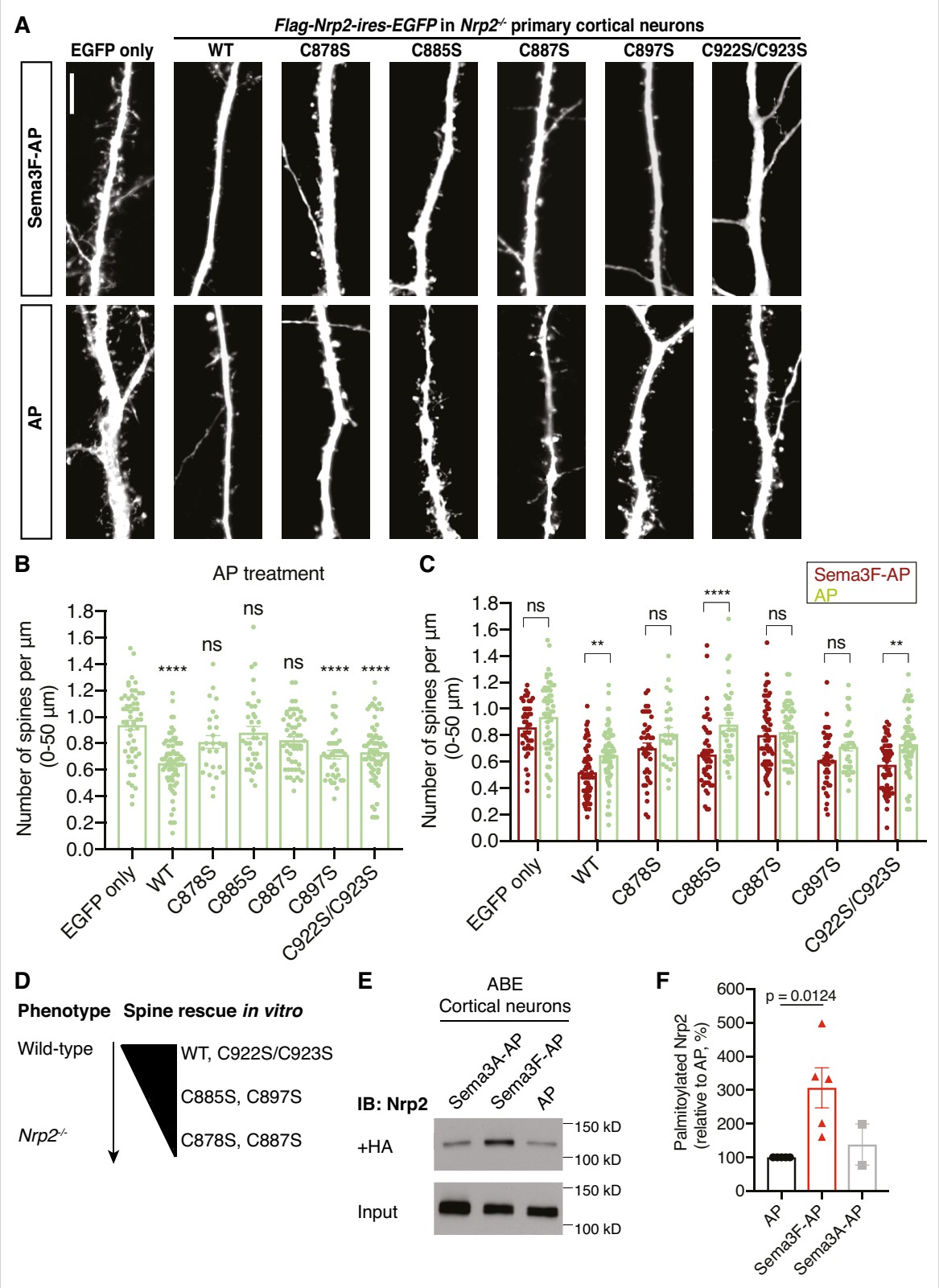

**Figure 4.** Select neuropilin-2 (Nrp2) cysteines are required for semaphorin 3F (Sema3F)/Nrp2-dependent dendritic spine pruning in deep layer primary cortical neurons. (**A–D**) Rescuing in vitro the *Nrp2⁻/⁻* dendritic spine phenotype to assess the role of individual Nrp2 cysteines in Sema3F/Nrp2-dependent dendritic spine collapse. E14.5 *Nrp2⁻/⁻* primary cortical neurons were transfected with various *flag-Nrp2-ires-EGFP* expression plasmids, including wild-type (*WT*) and cysteine-to-serine (*CS*) point mutants. At days in vitro (DIV)21 they were treated with 5 nM Sema3F-AP or 5 nM AP

*Figure 4 continued on next page*

*Figure 4 continued*

alone, for 6 hr, and next subjected to EGFP immunofluorescence to visualize neuronal architecture. (**A**) Representative images of neurons expressing the indicated plasmid and treated with either Sema3F-AP or AP. Images represent the 3D projection of a confocal stack. Scale bar in (**A**), 7 µm. (**B, C**) Quantification of dendritic spines, counted along the 0–50 µm apical dendritic segment and presented as number of spines per µm. Cumulative data from independent experiments are plotted in dot plots with mean ± SEM. The rescue ability of each Nrp2 protein is assessed in two ways: the ability of AP-treated neurons expressing each Nrp2 protein to constrain spines compared to AP-treated neurons expressing the backbone vector only (control) (**B**) and the ability of neurons expressing each Nrp2 protein to respond to Sema3F-AP compared to AP (control) (**C**) Graph in (**B**): One-way analysis of variance (ANOVA) (p<0.0001) followed by Bonferroni's test for multiple comparisons (compared to EGFP only: WT, p<0.0001; C878S, p=0.1215; C885S, p>0.9999; C887S, p=0.0628; C897S, p<0.0001; C922S/C923S, p<0.0001). Graph in (**C**): Two-way ANOVA (p<0.0001) followed by Sidak's test for multiple comparisons (Sema3F-AP vs. AP: EGFP only, p=0.4885; WT, p=0.0074; C878S, p=0.3555; C885S, p<0.0001; C887S, p=0.9985; C897S, p=0.3308; C922S/C923S, p=0.0017). (**D**) Graphic summary of the ability of Nrp2 proteins tested above to rescue the *Nrp2⁻ᐟ⁻* dendritic spine density phenotype in cortical neurons. WT and C922S/C923S Nrp2 proteins constrain dendritic spines, whereas certain Nrp2 CS mutants have either compromised (C885S, C897S) or abolished (C878S, C887S) rescue ability. Sema3F-AP treated: EGFP only, n=44; WT, n=63; C878S, n=39; C885S, n=46; C887S, n=60; C897S, n=38; C922S/C923S, n=53. AP treated: EGFP only, n=55; WT, n=67; C878S, n=26; C885S, n=36; C887S, n=53; C897S, n=36; C922S/C923S, n=64, where n is the number of neurons analyzed for each condition. (**E, F**) Sema3F enhances endogenous Nrp2 palmitoylation in cortical neuron cultures. ABE on DIV18 primary cortical neurons treated with either 5 nM Sema3F-AP, 5 nM AP (control), or 5 nM Sema3A-AP (additional control). (**E**) Immunoblots show palmitoylated (+HA) and input samples. -HA samples were processed in parallel with no evidence of non-specific signal (not shown). (**F**) Quantification of palmitoylated Nrp2 levels, calculated as the ratio of +HA to the respective input. Sema3F-AP and Sema3A-AP conditions are expressed relative to AP control (set at 100%). Cumulative data are plotted in dot plots with mean ± SEM. One-way ANOVA followed by Dunnett's test for multiple comparisons; Sema3A-AP vs. AP, p=0.8531 (AP and Sema3F-AP, n=5 experiments; Sema3A-AP, n=2 experiments).

The online version of this article includes the following source data and figure supplement(s) for figure 4:

**Source data 1.** Raw data for *Figure 4B and C*.

**Source data 2.** Raw, unedited blot from *Figure 4E*.

**Source data 3.** Raw, labeled blot from *Figure 4E*.

**Source data 4.** Raw data for *Figure 4F*.

**Figure supplement 1.** Semaphorin 3F (Sema3F)/neuropilin-2 (Nrp2)-dependent dendritic spine pruning in deep layer cortical neurons depends on distinct Nrp2 cysteine clusters.

**Figure supplement 1—source data 1.** Raw data for *Figure 4—figure supplement 1B*.

## Palmitoyltransferase ZDHHC15 palmitoylates Nrp2 and is required for Sema3F/Nrp2-dependent dendritic spine pruning in primary cortical neurons

To identify the DHHC enzymes that palmitoylate Nrp2, we carried out a Nrp2 palmitoylation screen in HEK293T cells by co-expressing exogenous wild-type Nrp2 and each of the 23 DHHCs (*Fukata et al., 2004*) and assessing palmitoylation with the ABE assay. Of the 23 DHHCs, ZDHHC15 (zinc finger DHHC domain-containing protein 15, also known as DHHC15) was among the three DHHCs that most strongly enhanced Nrp2 palmitoylation (data not shown). ZDHHC15, encoded by a gene located at the *Xq13.3* genetic locus in humans, is known to palmitoylate PSD-95 (*Fukata et al., 2004*), is strongly expressed in the mouse cerebral cortex (*Mejias et al., 2021*) and is enriched in Golgi membranes (*Ohno et al., 2006*). Therefore, we examined the role of ZDHHC15 in Nrp2 palmitoylation and function in cortical neurons.

We employed a loss-of-function approach using a *Zdhhc15* knockout (*Zdhhc15*-KO) mouse line (*Mejias et al., 2021*), exploring whether Nrp2 is a ZDHHC15 substrate by performing ABE in wild-type and *Zdhhc15*-KO primary cortical neurons. In *Zdhhc15*-KO cortical neurons overall Nrp2 palmitoylation is substantially reduced (~50%) compared to that in wild-type neurons (*Figure 6A and B*). Unlike Nrp2, Nrp1 palmitoylation in *Zdhhc15*-KO neurons is comparable to that observed in wild-type neurons (*Figure 6A and C*). To test for DHHC-substrate specificity, we assessed Nrp2 palmitoylation in the previously characterized *Zdhhc8* knockout (*Zdhhc8⁻ᐟ⁻*) mouse line (*Mukai et al., 2015*; *Mukai et al., 2008*; *Mukai et al., 2004*), since ZDHHC8 did not enhance Nrp2 palmitoylation in our screen in 293T cells. Nrp2 palmitoylation levels in *Zdhhc8⁻ᐟ⁻* cortical neurons are very close to those observed in wild-type cortical neurons (*Figure 6—figure supplement 1A and B*) and, therefore, Nrp2 does not appear to be a palmitoyl substrate of ZDHHC8 in these neurons.

Next, we asked whether ZDHHC15 plays a role in Sema3F/Nrp2-dependent dendritic spine pruning. Wild-type and *Zdhhc15*-KO deep layer primary cortical neurons were cultured, transfected with an EGFP-expressing plasmid, and at DIV21 they were treated with 5 nM Sema3F-AP or 5 nM

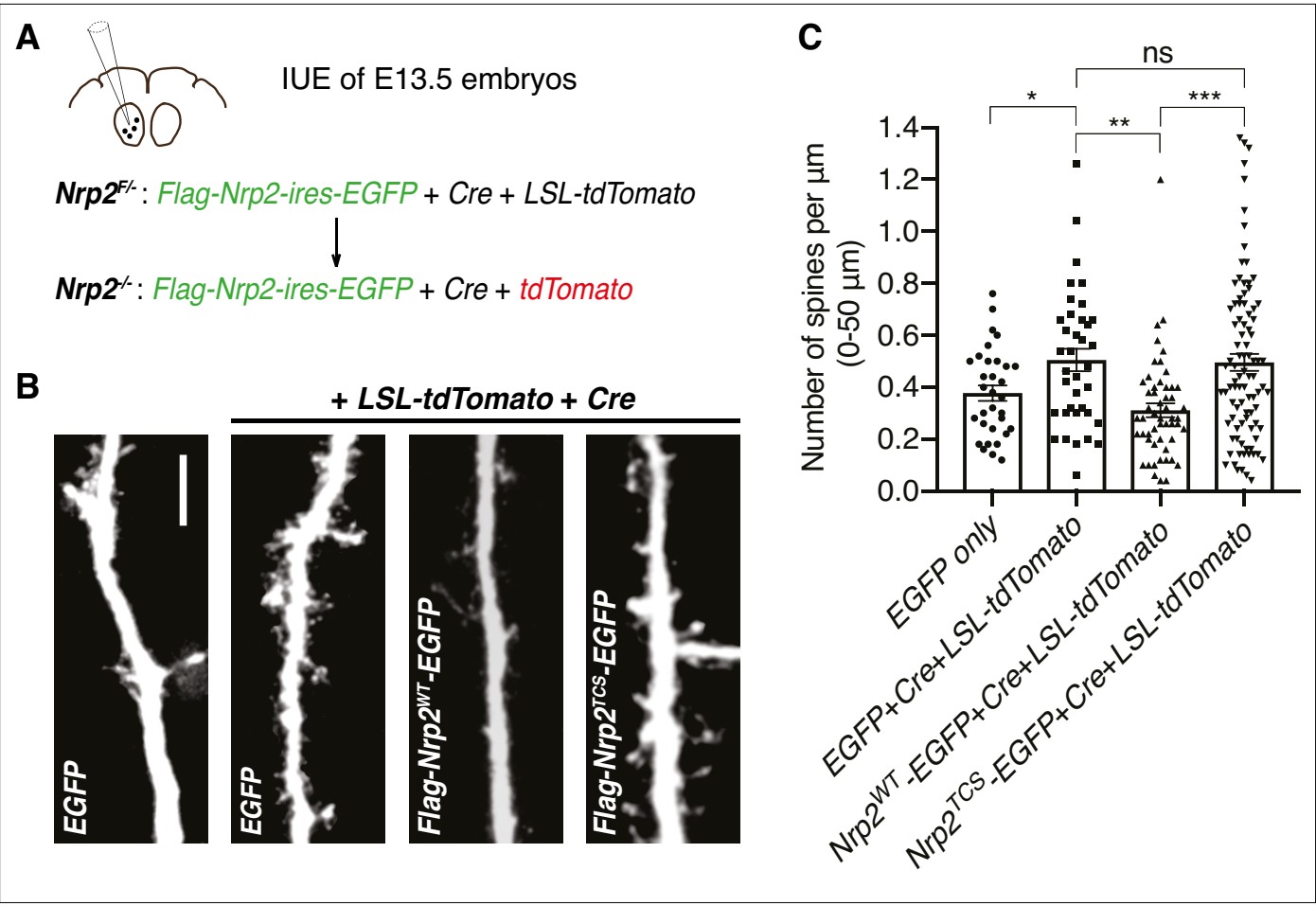

**Figure 5.** Transmembrane/juxtamembrane neuropilin-2 (Nrp2) cysteines are required in vivo for Nrp2-dependent dendritic spine pruning in deep layer cortical pyramidal neurons. (**A–C**) Rescuing in vivo the *Nrp2^-/-* dendritic spine phenotype to assess the role of membrane-associated Nrp2 cysteines in semaphorin 3F (Sema3F)/Nrp2-dependent dendritic spine retraction in layer V cortical pyramidal neurons in the mouse brain. (**A**) Schematic representation of the in utero electroporation (IUE) experimental approach. Embryonic day 13.5 (E13.5) *Nrp2^F/-* embryos were electroporated in utero either with EGFP or with [*EGFP+Cre+loxP-STOP-loxP-tdTomato* (*LSL-tdTomato*)] to excise the *floxed Nrp2* allele and render individual neurons *Nrp2^-/-*. *Floxed tdTomato* (*LSL-tdTomato*) serves as a reporter of Cre activity. These two electroporations serve as controls to assess the utility of this approach in reproducing the *Nrp2^-/-* spine density phenotype. Next, embryos were electroporated with either [*flag-Nrp2^WT-ires-EGFP+Cre+LSL-tdTomato*] or [*flag-Nrp2^TCS-ires-EGFP+Cre+LSL-tdTomato*] to assess the ability of wild-type (WT) or triple cysteine-to-serine (TCS) Nrp2 to constrain supernumerary dendritic spines resulting from Nrp2 deletion. All mice were analyzed on postnatal day 28 (**P28**). (**B**) Representative images of apical dendrites of electroporated layer V pyramidal neurons, which represent the 3D projection of a confocal stack, expressing the indicated combinations of plasmids. Scale bar, 7 µm. (**C**) Quantification of dendritic spines, counted along the proximal 50 µm (relative to the cell body) of the apical dendrite and presented as number of spines per µm. Cumulative data from IUE experiments (n≥3 mouse brains analyzed per scheme of injected plasmids) are plotted in scatter dot plots with mean ± SEM. One-way analysis of variance (ANOVA) (p=0.0001) followed by Tukey's test for multiple comparisons; *EGFP only* vs. [*EGFP+Cre+LSL-tdTomato*], p=0.0204; [*EGFP+Cre+LSL-tdTomato*] vs. [*flag-Nrp2^WT-ires-EGFP+Cre+LSL-tdTomato*], p=0.0032; [*EGFP+Cre+LSL-tdTomato*] vs. [*flag-Nrp2^TCS-ires-EGFP+Cre+LSL-tdTomato*], p=0.9975; [*flag-Nrp2^WT-ires-EGFP+Cre+LSL-tdTomato*] vs. [*flag-Nrp2^TCS-ires-EGFP+Cre+LSL-tdTomato*], p=0.0003. *EGFP only*, 32 neurons; [*EGFP +Cre+LSL-tdTomato*], 37 neurons; [*flag-Nrp2^WT-ires-EGFP+Cre+LSL-tdTomato*], 53 neurons; [*flag-Nrp2^TCS-ires-EGFP+Cre+LSL-tdTomato*], 92 neurons.

The online version of this article includes the following source data for figure 5:

**Source data 1.** Raw data for *Figure 5C*.

AP for 6 hr. They were then subjected to EGFP immunofluorescence, imaged and spine density was quantified along the largest dendrite. Unlike wild-type cortical neurons, which displayed a significant reduction in spine density following treatment with Sema3F-AP compared to AP (*Figure 6D and E*), *Zdhhc15*-KO neurons failed to exhibit apical dendritic spine pruning following Sema3F-AP treatment (*Figure 6D and F*).

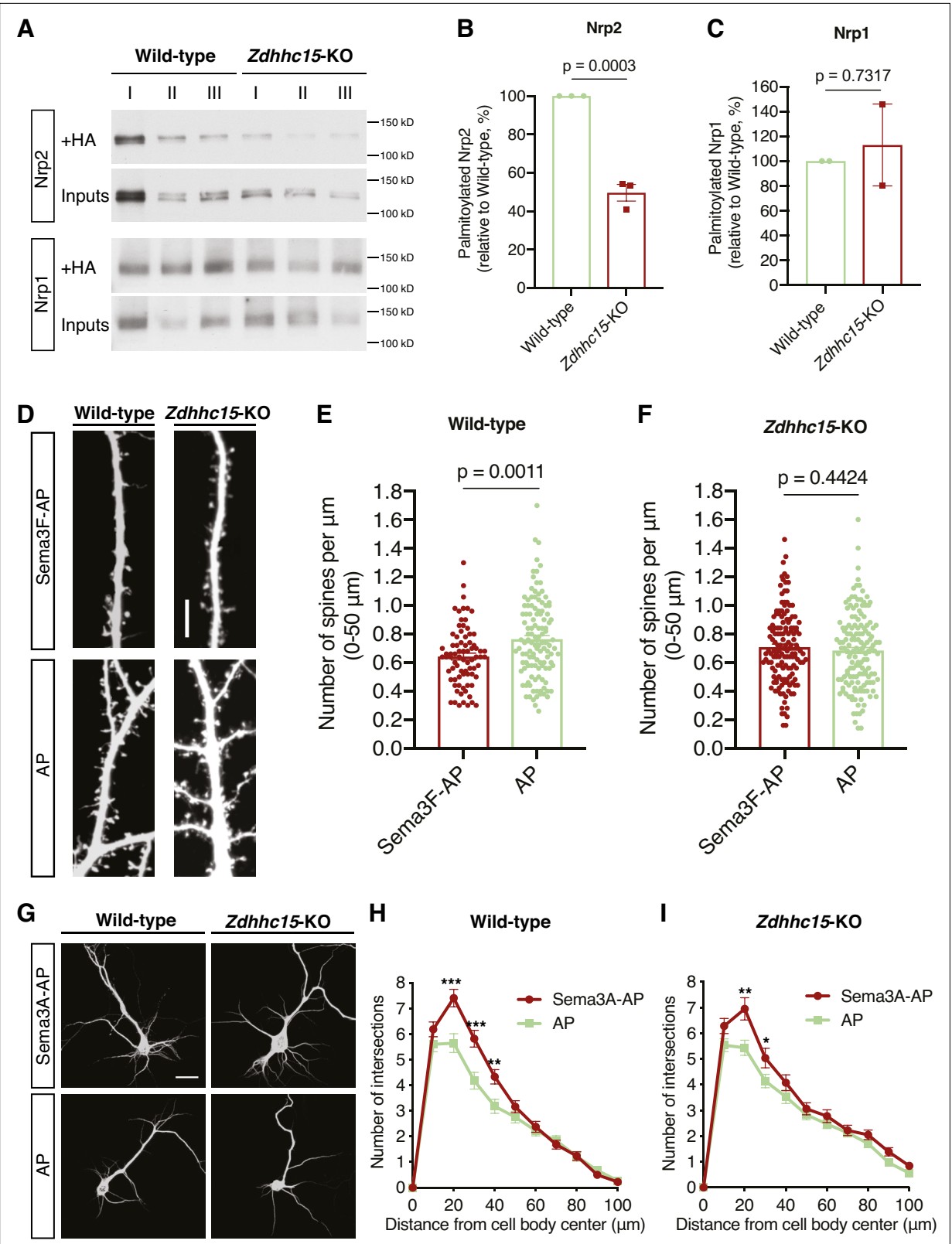

**Figure 6.** ZDHHC15 regulates neuropilin-2 (Nrp2) palmitoylation and function, but not Nrp1, in deep layer primary cortical neurons. (A–C) Acyl-biotin exchange (ABE) performed on embryonic day 14.5 (E14.5) days in vitro (DIV)12 wild-type (WT) C57BL/6J or *Zdhhc15*-KO primary cortical neurons. (A) Representative immunoblots of palmitoylated (+HA) and input samples for Nrp2 and Nrp1. (B, C) Quantification of palmitoylated Nrp2 (B) or Nrp1 (C), calculated as the ratio of palmitoylated protein (+HA) to the respective input; this ratio is set at 100% for WT neurons and the ratio for *Zdhhc15-*

*Figure 6 continued on next page*

*Figure 6 continued*

KO neurons is expressed as a percentage of the WT. In each ABE experiment, three different biological samples for each genotype (I, II, III are derived from different cortical cultures/mice and represent biological replicas) were processed in parallel and averaged in order to control for intraexperimental reproducibility and for potential differences in endogenous protein expression among samples. Nrp2, n=3 ABE experiments; Nrp1, n=2 ABE experiments. Cumulative data are plotted in scatter dot plots with mean ± SEM. Two-tailed t test; Nrp2 mean: WT = 100%, *Zdhhc15*-KO=49.64%; Nrp1 mean: WT = 100%, *Zdhhc15*-KO=113%. (**D–F**) Assessment of the ability of WT C57BL/6J or *Zdhhc15*-KO primary cortical neurons to respond to semaphorin 3F (Sema3F) by excess dendritic spine pruning. Neurons were transfected with an EGFP-expressing plasmid, and at DIV21 they were treated with 5 nM Sema3F-AP or 5 nM AP (control) for 6 hr, followed by EGFP immunofluorescence to visualize neuronal morphology. (**D**) Representative images are shown for WT and *Zdhhc15*-KO neurons. Scale bar, 7 μm. (**E, F**) Quantification of dendritic spines, counted along the proximal 50 μm (from cell body) on the largest dendrite, presented as number of spines per μm and plotted in scatter dot plots including mean ± SEM. Note that WT neurons exhibit significant Sema3F-induced spine retraction, whereas *Zdhhc15*-KO neurons invariably display no response to Sema3F-AP (compared to AP). Two-tailed t test; WT: n=3 experiments; Sema3F-AP, 78 neurons; AP, 119 neurons. *Zdhhc15*-KO: n=3 experiments; Sema3F-AP, 144 neurons; AP, 147 neurons. (**G–I**) Assessment of the ability of WT C57BL/6J or *Zdhhc15*-KO primary cortical neurons to respond to Sema3A by elaborating their dendritic tree. E14.5 primary cortical neurons treated at DIV12 with 5 nM Sema3A-AP or 5 nM AP (control) for 6 hr and subjected to microtubule-associated protein 2 (MAP2) immunofluorescence. (**G**) Representative images are shown for each genotype and represent the 3D projection of a confocal stack. Scale bar, 20 μm. (**H, I**) Quantitative assessment of dendritic arborization with Sholl analysis, presented as number of intersections at various distances from the center of the cell body, and plotted as mean ± SEM for each distance. Both WT and *Zdhhc15*-KO cortical neurons exhibit a more elaborate perisomatic dendritic arbor following Sema3A-AP treatment compared to AP (control) treatment. WT: n=2 experiments; Sema3A-AP, 72 neurons; AP, 62 neurons. *Zdhhc15*-KO: n=3 experiments; Sema3A-AP, 77 neurons; AP, 110 neurons. Multiple t tests: (**H**), ***p<0.001, **p=0.0043; (**I**), **p=0.003, *p=0.046.

The online version of this article includes the following source data and figure supplement(s) for figure 6:

**Source data 1.** Raw, unedited blot from *Figure 6A*.

**Source data 2.** Raw, unedited blot from *Figure 6A*.

**Source data 3.** Raw, labeled blot from *Figure 6A*.

**Source data 4.** Raw, labeled blot from *Figure 6A*.

**Source data 5.** Raw data for *Figure 6B*.

**Source data 6.** Raw data for *Figure 6C*.

**Source data 7.** Raw data for *Figure 6E*.

**Source data 8.** Raw data for *Figure 6F*.

**Source data 9.** Raw data for *Figure 6H*.

**Source data 10.** Raw data for *Figure 6I*.

**Figure supplement 1.** Neuropilin-2 (Nrp2) is not a substrate of palmitoyl acyltransferase ZDHHC8 in deep layer primary cortical neurons.

**Figure supplement 1—source data 1.** Raw, unedited blot from *Figure 6—figure supplement 1A*.

**Figure supplement 1—source data 2.** Raw, labeled blot from *Figure 6—figure supplement 1A*.

**Figure supplement 1—source data 3.** Raw data for *Figure 6—figure supplement 1B*.

**Figure supplement 1—source data 4.** Raw data for *Figure 6—figure supplement 1D*.

To address the specificity of these spine morphology effects, we tested the ability of cortical neurons lacking ZDHHC8, which plays important roles in the nervous system (*Mukai et al., 2015*; *Mukai et al., 2008*), to exhibit spine pruning in response to Sema3F. *Zdhhc8⁻/⁻* cortical neuron cultures were transfected with an EGFP expression plasmid, and at DIV21 they were treated with Sema3F-AP or AP for 6 hr followed by EGFP immunofluorescence. *Zdhhc8⁻/⁻* cortical neurons responded to Sema3F with significant spine retraction (*Figure 6—figure supplement 1C and D*), similar to wild-type cortical neurons.

Since we observed no effect of *Zdhhc15* loss-of-function on Nrp1 palmitoylation, we investigated whether ZDHHC15 is required for Sema3A/Nrp1-dependent basal dendritic elaboration (*Gu et al., 2003*). Wild-type and *Zdhhc15*-KO cortical neuron cultures were treated at DIV12 with 5 nM Sema3A-AP or 5 nM AP for 6 hr. Next, they were subjected to immunofluorescence with an antibody directed against the dendritic marker microtubule-associated protein 2 (MAP2) to visualize dendritic trees, imaged and subjected to Sholl analysis. Wild-type neurons exhibited enhanced elaboration of perisomatic dendrites in response to Sema3A-AP, compared to the AP control (*Figure 6G and H*), in line with the effect of Sema3A/Nrp1 signaling on basal dendrite elaboration of deep layer cortical pyramidal neurons in the mouse brain (*Gu et al., 2003*). Likewise, *Zdhhc15*-KO cortical neurons responded to Sema3A with enhanced elaboration of their basal dendrite arbors following Sema3A-AP treatment as compared to the AP control (*Figure 6G and I*).

These biochemical and phenotypic experiments in vitro reveal that ZDHHC15 plays essential roles in Nrp2 palmitoylation and Sema3F/Nrp2-dependent dendritic spine pruning, and that ZDHHC15 is dispensable for Nrp1 palmitoylation and Sema3A/Nrp1-dependent basal dendritic elaboration in primary deep layer cortical neurons. These results underscore the exquisite specificity among PATs for their neuronal substrates.

## ZDHHC15 controls Nrp2-dependent dendritic spine pruning in deep layer cortical pyramidal neurons in vivo

Since we observe that ZDHHC15 is required for Nrp2 palmitoylation and Sema3F/Nrp2-dependent dendritic spine pruning in deep layer cortical neurons in culture, we asked whether ZDHHC15 is also required for Nrp2-mediated functions in cortical pyramidal neurons in the mouse neocortex. We examined the dendritic spine density on the apical dendrite of layer V cortical pyramidal neurons in *Zdhhc15*-KO and wild-type mice using two different labeling techniques, a genetic labeling approach using the *Thy1-EGFP-m* mouse line that labels layer V cortical pyramidal neurons and the Golgi staining technique for sparse neuronal labeling. Our analysis of apical dendritic spines in wild-type and *Zdhhc15*-KO layer V cortical pyramidal neurons labeled with the *Thy1-EGFP-m* line shows that *Zdhhc15*-KO apical dendrites harbor more dendritic spines compared to wild-type neurons (*Figure 7A and B*), phenocopying previously observed effects of *Nrp2* loss-of-function in this same neuronal population in vivo (*Tran et al., 2009*). Analysis of Golgi-stained brains yielded similar results (*Figure 7—figure supplement 1A and B*). Next, given the lack of an effect of ZDHHC15 on Nrp1 palmitoylation and function, we assessed the Sema3A/Nrp1-dependent basal dendritic arbor elaboration in Golgi-stained wild-type and *Zdhhc15*-KO brains and observed no difference in dendritic arbor complexity between wild-type and *Zdhhc15*-KO layer V cortical pyramidal neurons (*Figure 7C and D*).

We next examined whether ZDHHC15 and Nrp2 exert their effects in the same signaling pathway in layer V cortical pyramidal neurons in vivo. Despite potential issues related to haploinsufficiency (dose-dependent effects) and/or random X-inactivation, we performed a genetic interaction experiment involving *Nrp2* and *Zdhhc15*. For this approach we analyzed littermates of three different genotypes: (1) *Nrp2*$^{+/-}$*;Zdhhc15*$^{+/+}$*;Thy1-EGFP,* (2) *Nrp2*$^{+/+}$*;Zdhhc15*$^{+/-}$*;Thy1-EGFP,* and (3) *Nrp2*$^{+/-}$*;Zdhhc15*$^{+/-}$*;Thy1-EGFP*. Our sample size for each genotype was small (n=2 mice/genotype) due to the difficulty in obtaining all four alleles in the three genetic configurations along with Thy1-GFP expression in littermates. Despite the shortcomings of this experiment, we observed that individual neurons in transheterozygous *Nrp2* and *Zdhhc15* mutant mice display a higher density of dendritic spines on the apical dendrite of deep layer cortical pyramidal neurons compared to single *Nrp2* or *Zdhhc15* heterozygotes (*Figure 7—figure supplement 1C and D*). These observations are in line with our biochemical and phenotypic experiments presented here and support the hypothesis that Nrp2 and ZDHHC15 function in the same signaling pathway in vivo.

We also examined whether ZDHHC15 contributes to Nrp2-dependent developmental guidance of fiber tracts, including the anterior commissure (AC) and select cranial nerves (*Giger et al., 2000*). Analysis of wild-type and *Zdhhc15*-KO brains using neurofilament staining revealed an intact AC in *Zdhhc15*-deficient mice (*Figure 7—figure supplement 1E*), with none of the hallmark AC defects known to be present in *Nrp2* mutants (*Chen et al., 2000*; *Giger et al., 2000*). Likewise, the examination of *Zdhhc15*-KO embryos with whole-mount neurofilament staining revealed normal development and guidance of cranial nerves (data not shown).

Taken together, our in vitro and in vivo data suggest a critical role for select palmitoylated Nrp2 cysteines in Nrp2 subcellular localization and function in deep layer cortical neurons. Further, they reveal that PAT-substrate specificity drives diversification of Nrp2 and Nrp1 palmitoyl substrates to control deep layer cortical pyramidal neuron morphology and, presumably, neural circuit assembly.

## Discussion

We show here that Nrp2 and Nrp1 are novel neuronal palmitoylation substrates and that select palmitoyl acceptor Nrp2 cysteines are critical for Nrp2 protein subcellular localization, for plasma membrane clustering and for Sema3F/Nrp2-dependent apical dendritic spine pruning in deep layer cortical pyramidal neurons in vitro and in vivo (*Figure 7E and F*). Importantly, a comparative analysis

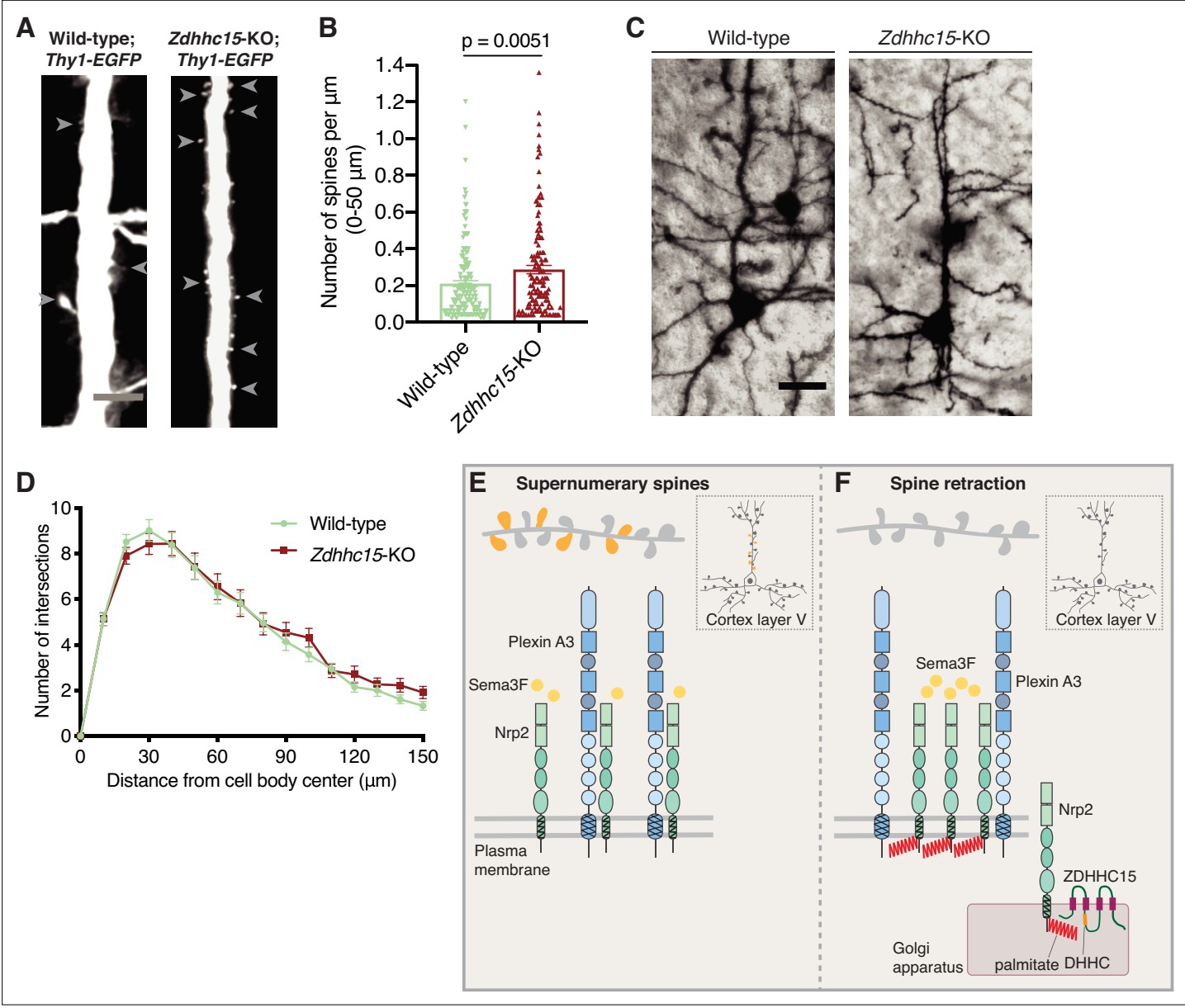

**Figure 7.** Selective effects of palmitoyl acyltransferase ZDHHC15 on layer V cortical pyramidal neuron morphology. (**A, B**) Assessment of dendritic spine density on the apical dendrite of layer V cortical pyramidal neurons in postnatal day 28 (**P28**) brains of wild-type;*Thy1-EGFP* and *Zdhhc15*-KO;*Thy1-EGFP* littermates. (**A**) Panels show representative images of the apical dendrite of Thy1-EGFP-labeled layer V cortical pyramidal neurons for each genotype following EGFP immunofluorescence. Scale bar, 7 μm. (**B**) Quantification of dendritic spines, counted along the proximal 50 μm (relative to the cell body) of the apical dendrite, presented as number of spines per μm, and plotted in scatter dot plots including mean ± SEM. Two-tailed t test; n=4 pairs of mice-littermates, wild-type vs. *Zdhhc15-KO*. Wild-type;*Thy1-EGFP*: 161 neurons, *Zdhhc15*-KO;*Thy1-EGFP*: 139 neurons. (**C, D**) Assessment of dendritic arborization of layer V cortical pyramidal neurons in wild-type and *Zdhhc15*-KO brains labeled using Golgi staining. (**C**) Representative images are shown for each genotype and are the 3D projection of a confocal stack. Scale bar, 30 μm. (**D**) Quantification of dendritic arborization using Sholl analysis reveals no significant difference in dendritic arbor complexity between wild-type and *Zdhhc15*-KO layer V pyramidal neurons of the cerebral cortex. Cumulative data are presented as number of intersections at various distances from the cell body, and are plotted as mean ± SEM. Multiple t tests; n=3 pairs of mice-littermates, wild-type vs. *Zdhhc15-KO*; wild-type: 72 neurons, *Zdhhc15*-KO: 64 neurons. (**E, F**) Schematic illustration of the effects of neuropilin-2 (Nrp2) palmitoylation on Nrp2 subcellular localization, surface distribution, and semaphorin 3F (Sema3F)/Nrp2-dependent dendritic spine pruning in deep layer cortical pyramidal neurons. (**E**) Unpalmitoylated Nrp2 is diffusely distributed over the entire plasma membrane and unable to constrain excess dendritic spines (depicted in dark yellow) along the apical dendrite of layer V cortical pyramidal neurons (the inset depicts a pyramidal neuron of the cerebral cortex). (**F**) Palmitoylation on transmembrane/juxtamembrane Nrp2 cysteines, mediated in part by ZDHHC15 in the Golgi apparatus and enhanced by Sema3F, enables Nrp2 clustering on distinct plasma membrane domains and Sema3F/Nrp2-dependent pruning of supernumerary dendritic spines on the apical dendrite of layer V pyramidal neurons of the cerebral cortex during postnatal development.

*Figure 7 continued on next page*

*Figure 7 continued*

The online version of this article includes the following source data and figure supplement(s) for figure 7:

**Source data 1.** Raw data for *Figure 7B*.

**Source data 2.** Raw data for *Figure 7D*.

**Figure supplement 1.** Selective effects of ZDHHC15 on neuropilin-2 (Nrp2)-dependent developmental processes in the mouse brain.

**Figure supplement 1—source data 1.** Raw data for *Figure 7—figure supplement 1B*.

**Figure supplement 1—source data 2.** Raw data for *Figure 7—figure supplement 1D*.

between Nrp2 and Nrp1 shows that PAT-substrate specificity contributes to the functional divergence and specification of Sema3F/Nrp2 as compared to Sema3A/Nrp1 signaling pathways in the CNS.

The discovery and functional characterization of neuropilin palmitoylation opens new avenues for investigating how these multifaceted neuronal cue receptors sculpt neuronal circuits. Despite Nrp2 and Nrp1 displaying similar palmitoylation patterns in their transmembrane and juxtamembrane segments, they exhibit critical differences with regard to their subcellular localization and function, and these can be partially ascribed to palmitoylation. Specifically, palmitoylation conveys upon Nrp2, a highly clustered localization pattern on the cell surface, whereas Nrp1 is diffusely localized on the plasma membrane regardless of its palmitoylation state. Further, our biochemical and phenotypic analyses of Sema3F/Nrp2-dependent cortical neuron dendritic spine pruning and Sema3A/Nrp1-dependent cortical neuron dendritic elaboration reveal that the PAT ZDHHC15 is required for Sema3F/Nrp2 function in cortical neurons, whereas it is not required for Sema3A/Nrp1 function in this same class of neuron. These observations show that palmitoylation endows Nrp2 and Nrp1 with distinct functional properties that allow these receptors to exert their well-defined and distinct effects on cortical layer V pyramidal neuron morphogenesis. This may be explained by palmitoylation events mediated by distinct PATs and/or palmitoylation at distinct subcellular sites being differentially important for protein localization and function. For example, palmitoylation close to the plasma membrane might regulate the targeting of a protein to dendritic spines and/or synapses, and these events may also include activity-dependent or ligand-dependent effects on protein trafficking (*Brigidi et al., 2014*; *Noritake et al., 2009*). Additional factors that may interact with palmitoylation signaling, including other posttranslational modifications (*Lin et al., 2009*; *Salaun et al., 2010*), sorting signals, and protein-protein interactions, may endow Nrp2 and Nrp1 with additional protein-specific localizing and functional properties.

Our finding that Nrp2 cysteines critically control Nrp2 localization, homo-multimerization, and function in cortical neurons raises the question as to whether these effects are mediated by palmitoylation per se, or they result from cysteine residue-dependent structural effects. Several lines of evidence presented here, including robust Nrp2 palmitoylation, the effect of 2-bromopalmitate on Nrp2 cell surface localization, the Sema3F-induced enhancement of Nrp2 palmitoylation, and the critical role of ZDHHC15 in Nrp2 function, provide strong support for a palmitoylation-dependent role for these cysteines. Interestingly, cysteine palmitoylation itself has been shown to impact protein dimerization (*Bhattacharyya et al., 2016*). Given its reversibility, palmitoylation could give rise to bidirectional transformation between a palmitoylated state and a non-palmitoylated state that takes part in disulfide-linked dimer formation. In addition, given the critical role of neuron-glial related cell adhesion molecule (NrCAM) in Sema3F-dependent dendritic spine pruning in cortical neurons (*Demyanenko et al., 2014*; *Duncan et al., 2021*), it will be of interest to investigate the role of Nrp2 palmitoylation in the physical and functional interactions between the Nrp2/PlexA3 holoreceptor complex and NrCAM.

We find that the C-terminal Nrp2 di-cysteine motif CCXXX* ('*' denotes a stop codon) does not appear to play essential roles in Nrp2 localization on Golgi membranes, cell surface compartmentalization, or dendritic spine pruning, though minor effects of these cysteines can be observed in certain localization assays. This C-terminal motif is also present in paralemmin (*Kutzleb et al., 1998*) and RhoB (*Adamson et al., 1992*), proteins in which the upstream cysteine is palmitoylated and the downstream cysteine is prenylated (*Adamson et al., 1992*; *Fukata and Fukata, 2010*; *Kutzleb et al., 1998*). It is likely that the Nrp2 C-terminal di-cysteine motif is important for other aspects of Nrp2 function such as axon guidance or axon pruning. Further, the similar cell surface distribution patterns of Nrp2$^{TCS}$ and Nrp2$^{Full\,CS}$ suggest that the juxtamembrane cysteines predominate over the C-terminal

cysteines in regulating Nrp2 subcellular localization. Even within the juxtamembrane cysteine cluster there appears to be synergistic and/or dominant effects among the three cysteines, as revealed by the non-additive effects of different cysteine mutants on overall Nrp2 palmitoylation and by the ability of these different Nrp2 proteins to regulate dendritic spine pruning. However, with regard to effects on overall protein clustering, more extensive Nrp2 CS mutants (e.g. Nrp2$^{TCS}$, Nrp2$^{Full\ CS}$) display more severe deficits in protein localization as compared to single CS mutants (*Figure 3—figure supplement 1* and data not shown). Overall, our observations favor a model whereby there is functional segregation within the Nrp2 transmembrane and cytoplasmic domains that is attributable in part to distinct roles in Nrp2 localization and function being imparted by distinct cysteine clusters, consistent with previous observations on palmitoylation effects on ionotropic receptors (*Hayashi et al., 2009*; *Hayashi et al., 2005*).

Our observation that Nrp2 distribution includes localization in the Golgi apparatus suggests that bulk Nrp2 palmitoylation occurs on Golgi membranes, and that palmitoylated Nrp2 enters the post-Golgi secretory pathway to undergo anterograde trafficking toward the plasma membrane. Interestingly, a significant fraction of Nrp2 protein is associated with Golgi membranes in dendrites of cortical neurons (*Figure 3C and D* and *Figure 3—figure supplement 3*), known as Golgi outposts (*Horton et al., 2005*). This suggests that Nrp2 may also be palmitoylated locally in response to stimuli including its ligand Sema3F, other signaling effectors or even neuronal activity, and that Nrp2 is then delivered to dendritic spines and synapses in response to these cues, as has been observed for other proteins (*Noritake et al., 2009*). Therefore, the enhancement we observe of Nrp2 palmitoylation by exposure to exogenous Sema3F (*Figure 4E and F*) may lead to rapid and/or controlled delivery of Nrp2 to synaptic sites. Moreover, Nrp2 and its ligand Sema3F are dynamically regulated in response to alterations in neuronal activity (*Lee et al., 2012*; *Wang et al., 2017*); in the case of Nrp2 this may in part be controlled by rapid cycles of Nrp2 palmitoylation-depalmitoylation in the vicinity of individual dendritic spines, a mechanism that can confer both spatial and temporal precision of protein localization and function (*Fivaz and Meyer, 2003*; *Fukata et al., 2004*). This idea is in line with studies demonstrating activity-dependent palmitoylation of synapse-associated proteins in neural tissue (*Brigidi et al., 2014*; *Hayashi et al., 2005*; *Noritake et al., 2009*).

The spatial and temporal organization of the palmitoylation network is largely governed by the precise spatiotemporal regulation of PATs (*Greaves and Chamberlain, 2011*; *Noritake et al., 2009*; *Ohno et al., 2006*), rendering substrate specificity a major determinant of the functional specification of cues that can regulate neuronal architecture and polarity. However, the landscape of protein palmitoylation is further enriched by modulators of DHHC activity and the concurrent activity of depalmitoylating enzymes (*Salaun et al., 2020*; *Tortosa et al., 2017*; *Yokoi et al., 2016*). Of note, there is increasing evidence that palmitoylation deficits lead to altered neuronal excitability and aberrant neuronal phenotypes linked to neural diseases (*Mansouri et al., 2005*; *Milnerwood et al., 2013*; *Mukai et al., 2015*; *Mukai et al., 2008*; *Mukai et al., 2004*; *Pinner et al., 2016*; *Raymond et al., 2007*; *Singaraja et al., 2011*; *Sutton et al., 2013*). In particular ZDHHC15, shown here to play essential roles in Nrp2 palmitoylation and function, has been implicated in X-linked intellectual disability (*Lewis et al., 2021*; *Mansouri et al., 2005*), impairments in learning and memory (*Wang et al., 2015*), hyperactivity associated with a novel environment and sensitivity to psychostimulants (*Mejias et al., 2021*), and also dendritic outgrowth and formation of mature spines in hippocampal neurons in vitro (*Shah et al., 2019*). Intriguingly, Sema3F/Nrp2 signaling has been shown to play important roles in hippocampal circuit function, and its dysregulation may cause epileptic activity in mice (*Eisenberg et al., 2021*; *Li et al., 2022*; *Sahay et al., 2005*). Moreover, neuropilins are key players in cancer growth and progression and response to antineoplastic therapies (*Napolitano and Tamagnone, 2019*; *Rizzolio and Tamagnone, 2011*), and they also play important roles in the function of the immune system (*Kumanogoh and Kikutani, 2013*) and the vascular system (*Gu et al., 2003*; *Simons et al., 2016*). Our experiments identify neuropilins as new members of the neuronal palmitoyl proteome, reveal new DHHC enzymatic substrates in the CNS, and define DHHC enzyme-substrate specificity as a novel mechanism specifying the functional identity of neuronal substrates. These results advance our understanding of CNS development and function and may prove invaluable for the development of targeted therapeutic approaches directed toward amelioration of neural disorders associated with aberrant function of palmitoylation signaling pathways.

# Materials and methods

## Animals

Animal procedures were carried out in conformity with the policies and guidelines of the Animal Care and Use Committee (ACUC) of the Johns Hopkins University (protocol # MO20M48), which are established according to the US National Research Council's Guide to the Care and Use of Laboratory Animals and in compliance with the Animal Welfare Act and Public Health Service Policy. Mice were handled with care and every effort was made to minimize suffering. Wild-type C57BL/6J mice and *Thy1-EGFP-m* transgenic mice were purchased from the Jackson Laboratory. The *Zdhhc15* knockout (*Zdhhc15*-KO) mouse line was generated in Dr Tao Wang's laboratory (Johns Hopkins University, Baltimore, MD, USA) (*Mejias et al., 2021*). The *Zdhhc8* knockout mouse line was provided to our lab by Dr Joseph Gogos (Columbia University, New York, NY, USA) and has been characterized (*Mukai et al., 2015*; *Mukai et al., 2008*; *Mukai et al., 2004*).

## Cell lines

All cell lines used in this study were obtained from a commercial source and meet the characterization, authentication, and safety standards according to the distributor (American Type Culture Collection-ATCC): HEK293T cells (ATCC, Cat no. CRL-11268 RRID:CVCL_1926), COS-7 cells (ATCC, Cat no. CRL-1651, RRID:CVCL_0224), and Neuroblastoma 2a (Neuro-2a) cells (ATCC, Cat no. CCL-131 RRID:CVCL_0470). Neuro-2a cells were authenticated by ATCC using short tandem repeat (STR) profiling. HEK293T cells were authenticated by the Genetic Resources Core Facility (GRCF) of the Johns Hopkins University by STR profiling. Authentication of COS-7 cells was not performed since STR profiling cannot be provided by any commercial facility. However, verification of COS-7 cells was confirmed by morphological examination and stereotypic behavior of these cells in a variety of cell collapse and protein localization assays, as observed previously in the field. Mycoplasma testing for all three cell lines mentioned above was performed at the Johns Hopkins GRCF, and results were negative for Mycoplasma contamination for all cell lines. The cell line authentication STR profile reports and Mycoplasma testing reports are available at this study's data archiving database (https://doi.org/10.7281/T1/OY2X8T). Cells in culture were maintained in culture media consisting of DMEM (Dulbecco's Modified Eagle Medium), fetal bovine serum (FBS, 10% final), penicillin and streptomycin (50 U/ml), and Glutamax supplement (1× final, Thermo Fisher Scientific), in a humidified incubator at 37°C with 5% $CO_2$. Cells were plated in six-well dishes for biochemical experiments or in 12-well or 24-well dishes on glass coverslips for immunofluorescence, at the desired confluency. Cells were transfected using Lipofectamine 2000 reagent (Thermo Fisher Scientific), according to the manufacturer's instructions, and processed 24–48 hr after transfection based on protein expression.

## Primary cortical neuronal cultures

Timed-pregnant female mice were either obtained from external organizations or generated in-house by plug checks. For deep layer cortical neuronal cultures the hemispheres (excluding ventral structures and the olfactory bulb) were dissected out from embryos of both sexes from timed-pregnant mice at E14.5. During dissection the dissected tissue was kept on ice-cold Leibovitz's L-15 medium (Thermo Fisher Scientific). Tissue was digested in HBSS (Hank's balanced salt solution) containing 0.1% trypsin, in a 37°C-water bath for 15 min. Next, cortices were washed twice with HBSS containing 10% FBS to inactivate trypsin and dissociated in neuron growth medium containing 10% FBS by gently passing them several times through a glass Pasteur pipette. Dissociated neurons were plated onto six-well dishes for biochemical experiments or onto 12-well or 24-well dishes on glass coverslips for immunofluorescence, at the desired confluency. Next day, medium was replaced by fresh neuron growth medium and thereafter half of the medium was changed every 1 or 2 days. Neuron growth medium consisted of Neurobasal medium (Thermo Fisher Scientific, Gibco) supplemented with 2% B-27 supplement (Gibco), 2 mM Glutamax (Gibco), and 1× penicillin/streptomycin (Gibco). Neuronal cultures were maintained in a humidified incubator at 37°C with 5% $CO_2$.

Dish/coverslip preparation: Round glass coverslips were treated with nitric acid overnight, next washed with $ddH_2O$ and ethanol, and stored in 95% ethanol. The day of culture, dishes, with or without coverslips, were coated with 0.1 mg/ml poly-D-lysine (Sigma) diluted in $ddH_2O$ at 37°C for at least 3 hr. Before plating, poly-D-lysine was removed and dishes were washed twice with DPBS (Dulbecco's phosphate-buffered saline [PBS]).

## Plasmids

Plasmids were generated with standard cloning techniques. Briefly, polymerase chain reaction-amplified inserts carrying the appropriate restriction sites on the 5' and 3' ends were digested and ligated with the backbone vector. N-terminally flag-tagged Nrp1 and Nrp2 expression plasmids were generated in a *pCIG2-ires-EGFP* backbone vector containing the preprotrypsin signal peptide followed by the flag epitope tag and the Nrp1 or Nrp2 coding sequences in frame with the above elements. N-terminally pHluorin-tagged Nrp2 expression plasmid was generated by subcloning in frame the endogenous signal peptide of Nrp2, the pHluorin coding sequence and the Nrp2 coding sequence (downstream of its signal peptide) in a *pCAGGS-ires-dsRED* backbone vector. The GFP variant pHluorin displays strong fluorescence (bright green) at neutral pH, which occurs upon receptor surface localization, whereas it does not fluoresce in vesicles (pH <6.0). As such, it provides a robust assay for cell surface protein visualization in live cells, while upon regular EGFP immunofluorescence on permeabilized cells allows total (surface and intracellular) protein visualization. CS point mutants of Nrp1 and Nrp2 were generated by amplification of the wild-type protein with reverse primers harboring the desired muta-tion(s). All clones were fully sequenced and protein expression was assessed by western blotting in heterologous cells and immunofluorescence in cell lines and cultured neurons. The CS point mutations did not alter the molecular weight of Nrp2 or Nrp1, as evidenced by western blotting.

## Transfections

Transfection of COS-7 cells, 293T cells, and primary cortical neurons was performed with Lipofect-amine 2000 (Thermo Fisher Scientific, Cat no. 11668-019) according to the manufacturer's guide. In brief, the appropriate amounts of DNA and Lipofectamine were added in separate tubes containing Opti-MEM I medium and incubated at room temperature for 5 min. Next, diluted DNA and diluted Lipofectamine were mixed and co-incubated at room temperature for 20 min. The mix was added to cells and after ~5 hr the medium was replaced by fresh medium. Primary cortical neurons were trans-fected between DIV7 and DIV9. Transfection of Neuro-2a cells was performed with Metafectene Pro (Biontex, Germany) according to the manufacturer's instructions.

## 2-Bromopalmitate experiments

2-Bromopalmitate (or 2-bromohexadecanoic acid, Sigma-Aldrich, Cat no. 238422) was dissolved in dimethyl sulfoxide (Sigma-Aldrich, Cat no. D2650) for making concentrated stock solution. Stock solution was diluted in the appropriate culture medium to a final concentration of 10 µM. Cells were incubated with either 2-bromopalmitate-containing medium or control medium (the same quantity of solvent diluted in culture medium) overnight, in a humidified chamber at 37°C with 5% $CO_2$. The use of ethanol as a solvent worked equally well. No particular toxicity on cultured cells was observed following incubation.

## Acyl-biotin exchange (ABE) assay

ABE was performed according to published protocols (*Drisdel and Green, 2004*; *Wan et al., 2007*), with slight modifications as reported in the study of *Hayashi et al., 2009*. Briefly, the workflow of this assay involves four main steps: (1) Tissue is harvested and lysed in lysis buffer containing *S*-methyl methanethiosulfonate (Sigma, Cat no. 64306) to block free thiol groups (-SH). (2) Incubation with a buffer containing hydroxylamine ($NH_2OH$, Fisher Scientific, Cat no. 26103) that cleaves cysteine-linked palmitate thioester bonds or with control buffer that instead of hydroxylamine contains Tris. (3) Samples are incubated with a buffer containing biotin-HPDP (Soltec Ventures, Cat no. B106) that attaches to newly exposed cysteine thiols. Both buffers (for hydroxylamine-treated and control-treated samples) contain biotin-HPDP. Between steps 1–3, protein was precipitated by incubation with –20°C-cold 80% acetone at –20°C overnight and protein pellet was washed several times with the same solution before proceeding to next step. (4) Biotinylated proteins are affinity-purified with strepta-vidin agarose resin (Thermo Scientific, Cat no. 20349). The use of neutravidin agarose resin yielded the same results. ABE samples obtained with this assay were resolved with SDS-polyacrylamide gel electrophoresis (SDS-PAGE) and immunoblotting with protein-specific antibodies. In this assay, the proteins detected in the plus hydroxylamine (+HA) sample but not in the minus hydroxylamine (-HA) sample (which serves as an internal negative control) represent palmitoylated proteins and collectively comprise the palmitoyl proteome. In all ABE experiments, for each tissue sample, both +HA and -HA

samples were processed in parallel and the specificity of +HA signal was confirmed by the complete absence of signal in -HA sample in western blotting.

## Western blotting and quantification

Samples were mixed with Laemmli sample buffer (Laemmli, Bio-Rad, Cat no. 1610747), boiled at 95°C for 5 min, stored at –20°C and analyzed with SDS-PAGE and western blotting. Briefly, samples were loaded on 4–20% gradient mini-PROTEAN TGX precast protein gels (Bio-Rad, 10-well Cat no. 4561094, or 15-well Cat no. 4561096) and run under standard protein electrophoresis conditions. Proteins on gels were transferred on PVDF membrane (Immobilon-P, Millipore, Cat no. IPVH00010) and blocked with 5% non-fat dry milk (Scientific, Cat no. M0841), diluted in TBS-T (TBS-Tween 20), for 1 hr at room temperature. Next, membranes were incubated with primary antibody in 1% milk/ TBS-T, overnight at 4°C, with end-to-end rotation. Next day, membranes were washed with TBS-T three times and incubated with horseradish peroxidase (HRP)-conjugated species-specific secondary antibody in 1% milk, to a final concentration 1:10,000, for 1 hr at room temperature, with end-to-end rotation. Membranes were washed with TBS-T four times. Signal detection was performed with ECL Western Blotting Detection Reagent (GE Healthcare, Cat no. RPN2109) or ECL Prime Western Blotting Detection Reagent (GE Healthcare, Cat no. RPN2232) or Clarity Western ECL substrate (Bio-Rad, Cat no. 1705060). Different exposures were obtained for each experiment to ensure that the detected signals were in the linear range. Western blot densitometry was performed with the gel analysis tool of ImageJ. For ABE experiments, palmitoylated protein was calculated as the ratio of +HA (hydroxylamine) signal to the respective input and subsequently all values (ratios) were expressed as a percentage of the ratio for wild-type protein (as explained in figure legends).

## Surface staining in COS-7 cells and clustering analysis

### Surface staining

COS-7 cells were transfected with flag-tagged Nrp2 or Nrp1 expression plasmids. Next day, cells were incubated with anti-FLAG antibody (mouse monoclonal, Sigma, Cat no. F1804) diluted in culture medium at a final concentration 1:50, at 10°C for 20 min. Then, they were washed with fresh culture medium and fixed with ice-cold 4% paraformaldehyde in PBS for 10 min. Following fixation, cells were washed with PBS, blocked with 10% donkey serum and 0.1% Triton X-100 in PBS for 1 hr at room temperature, and incubated with fluorescent secondary antibody (1:1000) for 1 hr at room temperature. Following antibody incubation, cells were washed with PBS and coverslips with cells were mounted on microscope slides with mounting medium (Vectashield).

### Clustering analysis

Cells were imaged with the acquisition of single plane images. Protein clustering on the surface of COS-7 cells was quantified with particle analysis (ImageJ). Briefly, images were converted to 8-bit, thresholded and converted to binary. Next, the image was selected and a mask of the signal was created. The 'mask' image was analyzed with the 'Analyze Particles' tool, with protein particles (or clusters) defined based on two parameters, size in $\mu m^2$ (0-infinity) and circularity (0.00–1.00). Among the parameters provided in the results table for each cell analyzed are the particle count, the total area of fluorescence and the average size (area) of measured particles. Based on these, we calculate the ratio Count/Total Area, presented as 'number of particles per $\mu m^2$ of fluorescence area' (*Figures 1C, E and 3B* and *Figure 3—figure supplement 1B*), which represents the degree of protein clustering; the more punctate/clustered the protein, the higher the ratio. The mean size of particles (*Figure 3—figure supplement 1C*) is an alternative measure of protein clustering; a cell with punctate protein distribution gives a low average size of protein particles, while a cell with diffuse protein distribution gives a higher average size of protein particles. Although particle analysis provides an estimate of protein distribution, it tends to underestimate differences in protein localization observed by visual inspection because thresholding softens differences between small and large particles and distinct localization patterns between Nrp2 CS mutants are abated. Therefore, this approach has a low sensitivity for capturing protein localization differences observed in neurons that have a lot of thin processes and small cytoplasmic volumes; this is the reason why particle analysis is not shown in *Figure 3—figure supplement 2*.

## Immunofluorescence

### Immunofluorescence in vitro

Cells adherent on coverslips were fixed with ice-cold 4% paraformaldehyde for 10 min at room temperature, washed with PBS and blocked with 10% goat or donkey serum (depending on the primary antibody) and 0.1% Triton X-100 in PBS for 1 hr at room temperature. Next, cells were incubated with primary antibodies at 4°C overnight. Next day, they were washed with PBS and next incubated with secondary antibodies diluted 1:1000 for 1 hr at room temperature in the dark. Cells were washed four times, 10 min each, with 1× PBS and coverslips with cells were mounted on microscope slides with mounting medium (Vectashield, Vector Laboratories).

### Immunofluorescence in vivo

Mice were anesthetized and perfused transcardially with PHEM buffer (1× PHEM: 60 mM PIPES, 25 mM HEPES, 5 mM EGTA, 1 mM $MgSO_4$; pH = 6.9) followed by ice-cold PHEM buffer containing 4% paraformaldehyde and 3.7% sucrose. Brains were dissected and post-fixed with perfusion buffer (4% paraformaldehyde, 3.7% sucrose, in PHEM buffer) for 2 hr at 4°C followed by washes with PBS and incubation with 30% sucrose in PBS at 4°C overnight. Next day, brains were embedded in NEG-50 frozen section medium following regular procedures. Sectioning was performed with a Leica cryostat at the coronal or sagittal plane at 30 µm thickness. Sections were placed on precleaned superfrost plus microscope slides (Fisher Scientific), left dry at room temperature and stored at –80°C until staining. For immunofluorescence, slides were sealed with ImmEdge hydrophobic barrier Pen (Vector Laboratories, Cat no. H-4000) and sections were blocked with 10% goat or donkey serum and 0.1% Triton X-100 in PBS for 1 hr at room temperature followed by incubation with primary antibodies diluted in 1% serum and 0.1% Triton X-100 in PBS, at 4°C overnight. Tissue was washed with PBS followed by incubation with secondary antibodies for 1 hr at room temperature. Finally, tissue was washed with PBS and covered with mounting medium (Vectashield) and a rectangular coverslip.

### Neurofilament staining (floating slice immunofluorescence)

Mice were transcardially perfused with PBS followed by 4% paraformaldehyde in PBS. Brains were dissected and post-fixed with 4% paraformaldehyde, washed with PBS, and embedded in 3% low melting point agarose. Brains were sectioned with a vibratome at the coronal plane, at 150 µm thickness, and processed for immunofluorescence floating in PBS in culture dishes. Slices were incubated with permeabilization solution (PBS, $H_2O$, Bovine Serum Albumin, Triton X-100) at 4°C for at least 6 hr with gentle agitation. Next, slices were incubated with anti-neurofilament (2H3) antibody (mouse monoclonal, Developmental Studies Hybridoma Bank) diluted in permeabilization solution at 4°C overnight. Next, they were washed with PBS at room temperature and subsequently incubated with secondary antibody diluted in permeabilization solution supplemented with 5% normal goat serum at 4°C overnight, washed with PBS, and mounted on microscope slides with PBS. Excess PBS was removed and slices were left to dry for 30 min and subsequently covered with Fluoro Gel with DABCO (Electron Microscopy Sciences, Cat no. 17985) and a rectangular coverslip.

## Antibodies

### Primary antibodies (immunofluorescence, immunoprecipitation, western blotting)

Flag (mouse monoclonal, Sigma, Cat no. F1804); GFP (chicken IgY, Avés, Cat no. GFP-1020); dsRED (rabbit polyclonal, Living Colors, Clontech, Cat no. 632496); MAP2 (mouse monoclonal, Sigma, Cat no. M1406); GM-130 (rabbit monoclonal, Abcam, EP892Y, Cat no. ab52649); neuropilin-2 (Nrp2) (rabbit polyclonal, Cell Signaling, Cat no. 3366S); Nrp2 (goat polyclonal, Research & Development, Cat no. AF2215); PlexA3 (rabbit polyclonal, Abcam, Cat no. ab41564), neuropilin-1 (Nrp1) (rabbit, Abcam, Cat no. ab81321); Nrp1 (goat polyclonal, R&D, Cat no. AF566); PSD-95 (mouse, NeuroMab, Cat no. 75028); SAP102 (mouse monoclonal, NeuroMab, Cat no. 75-058); neurofilament (2H3) (mouse monoclonal, Developmental Studies Hybridoma Bank); Ctip2 (rat monoclonal, Abcam, [25B6], Cat no. ab18465).

### Secondary antibodies

Immunofluorescence: CF488A donkey anti-chicken IgY (H+L) (Biotium, Cat no. 20166); Alexa Fluor 488 goat anti-mouse IgG (Life Technologies, Cat no. A11001); Alexa Fluor 488 donkey anti-goat IgG (Cat no. A11055); Alexa Fluor 555 donkey anti-rabbit IgG (Life Technologies, Cat no. A-31572); Alexa Fluor 546 donkey anti-mouse IgG (Thermo Fisher Scientific, Cat no. A10036); Alexa Fluor 555 goat anti-rabbit IgG (Cat no. A-21428); DyLight 649 donkey anti-mouse IgG (Jackson ImmunoResearch); Alexa Fluor 405 goat anti-rabbit IgG (Thermo Fisher Scientific, Cat no. A-31556); Alexa Fluor 647 goat anti-chicken IgG (Thermo Fisher Scientific, Cat no. A-21449); Alexa Fluor 647 donkey anti-rabbit IgG (Thermo Fisher Scientific, Cat no. A-31573); Alexa Fluor 647 donkey anti-rat IgG (Abcam, Cat no. ab150155).

Western blotting: HRP-conjugated antibody directed against the host species of the primary antibody.

## Nrp2–Golgi association analysis

### Immunofluorescence

E14.5 *Nrp2$^{-/-}$* cortical neurons in culture were transfected with pHluorin-tagged Nrp2 expression plasmids and at DIV17 they were stained with a GFP antibody to detect Nrp2 (total, surface, and intracellular) and a GM130 antibody to visualize *cis*-Golgi. Neurons were imaged with an upright LSM700 confocal microscope (Zeiss) with the acquisition of single plane images. All imaging parameters were kept constant during data acquisition.

### Quantitative assessment of colocalization

Performed with ImageJ. The Nrp2 and Golgi (GM130) channels were split and thresholded creating binary images (a different threshold was used for each of the two channels, while thresholds were kept constant throughout the analysis). The area of Nrp2 signal ($A_{Nrp2}$) and Golgi signal ($A_{Golgi}$) was selected and measured ($\mu m^2$). The colocalization plugin was used to give the colocalized image that was used to create a mask that represents the colocalized area ($A_{Nrp2/Golgi}$), which was then selected and measured. The ratio of the colocalized area to the Nrp2 area ($A_{Nrp2/Golgi}/A_{Nrp2}$) represents the fraction of Nrp2 that is associated with Golgi. This ratio was further normalized to the total Golgi area ($A_{Golgi}$) to take into account differences in the abundance of Golgi membranes between neurons.

### Golgi isolation

Performed on wild-type mouse whole brain according to *Current Protocols in Cell Biology* (Unit 3.9, Basic Protocol 2). Mouse brain was harvested and gently homogenized with a Potter-Elvehjem pestle to avoid Golgi stack fragmentation. Golgi stacks were isolated by flotation through a discontinuous sucrose gradient. Briefly, light mitochondrial pellet was resuspended in 2 ml buffer and mixed with 8 ml 2.0 M sucrose so that final sucrose concentration is about 1.55 M. Five ml of this mix was transferred to a 17 ml ultracentrifuge tube (Beckman) and overlayed with the following sucrose solutions: 4 ml 1.33 M sucrose, 2 ml 1.2 M sucrose, 2 ml 1.1 M sucrose, 2 ml 0.77 M sucrose, 0.25 M sucrose to fill the tube. Gradients were centrifuged at 100,000 × *g* at 4°C for 1 hr. The following fractions were collected: 1.55M, 1.55M/1.33M, 1.33M, 1.1M/1.2M, intervening fraction, 0.77M/1.1M. Samples were prepared and stored for SDS-PAGE and immunoblotting.

## Live imaging

Live imaging was performed to visualize cell surface localization of pHluorin-tagged Nrp2 in primary cortical neurons or Neuro-2a cells. The pHluorin epitope tag serves as a robust fluorescent reporter of protein expressed on the cell surface because of its sensitivity to pH. Cells were cultured in individual glass bottom dishes (MatTek Life Sciences, Cat no. P35G-0-14C) precoated with poly-D-lysine and transfected with appropriate plasmids. One or two days after transfection, they were imaged live at an inverted LSM700 confocal microscope (Zeiss) with a 40× oil lens. Imaging parameters were set prior to the start of these experiments, and were kept constant during image acquisition.

## Fluorescence recovery after photobleaching (FRAP)

Wild-type cortical neurons were cultured and plated on glass bottom dishes precoated with poly-D-lysine. Between DIV5 and DIV8 neurons were transfected with the indicated *pHluorin-Nrp2-ires-dsRED* expression plasmids with the method of Lipofectamine. A few days later, they were imaged with an inverted LSM700 microscope (Zeiss) with an incubation chamber (PeCon) (temperature: 37°C, $CO_2$: 5%), using a 63× NA1.4 oil immersion lens. For the performance of FRAP, we used the following protocol: a ROI containing the main dendritic process was selected for FRAP analysis. Images were acquired every 30 s; five images were acquired prior to bleaching (prebleach time: 2.5 min) and 31 additional images were acquired post-bleaching (total post-bleach time: 15.5 min). Analysis: Fluorescence intensity of time-lapse images was determined by measuring the fluorescence intensity within the ROI and correcting for fluorescence decay and for background fluorescence. Imaging parameters were set prior to image acquisition and were kept constant throughout the course of these experiments.

## Immunoprecipitation in heterologous cells

Cell growth medium was aspirated and lysis buffer (TNE buffer: NP-40, Tris, EDTA, NaCl, protease inhibitor cocktail) was added in cells in the culture plate. Cells were scraped off, mechanically lysed on ice by passage through 1 ml syringes and incubated with rotation at 4°C for 1 hr. Samples were precleared with protein A/G agarose resin (Pierce, Thermo Fisher Scientific, Cat no. 20421) at 4°C for 2 hr. Next, after the resin was discarded, a small fraction of each sample was kept as input and the remaining samples were incubated with the appropriate antibody at 4°C overnight. Next, samples were incubated with protein A/G agarose with end-over-end rotation at 4°C for 2 hr. Resin was washed twice with high-salt buffer (500 mM NaCl) and twice with low-salt buffer (150 mM NaCl). Proteins were eluted from resin with Laemmli sample buffer diluted in TNE buffer, boiled at 95°C for 5 min and analyzed with SDS-PAGE and western blotting.

## Alkaline phosphatase-fused ligand production

Alkaline phosphatase (AP)-tagged Sema3A and Sema3F or AP expression plasmids were transfected into 293T cells with the method of Lipofectamine. Cells were allowed to secrete the ligand for 2–5 days. The culture supernatant was collected and transferred to a 50 ml MW filter tube (Centricon filters, Millipore) and was centrifuged at 35 rpm for 10 min at 4°C. A 50 K filter was used for AP concentration and a 100 K filter was used for concentration of Sema3A-AP or Sema3F-AP. The liquid above the filter was transferred to a new tube and ligand concentration was determined with a spectrophotometer by measuring AP activity by the change in absorbance at OD 405 nm. This figure was then converted to nM. The concentrated ligand was aliquoted and stored at –80°C. For each experiment, a new aliquot was thawed and used for culture treatment to avoid protein decay from thawing and refreezing.

## *Nrp2⁻/⁻* rescue experiments

For in vitro rescue, E14.5 *Nrp2⁻/⁻* primary cortical neurons were transfected at DIV7 with the indicated plasmids. At DIV21 neurons were treated with 5 nM Sema3F-AP or AP for 6 hr and afterward they were processed for immunofluorescence. Confocal images (stacks) were taken with an LSM700 with a 63× oil lens. Analysis was performed with ImageJ and dendritic spines were counted along the 0–50 µm apical dendritic segment from the cell body. For in vivo rescue, IUE was performed on E13.5 *Nrp2^F/-* embryos of timed-pregnant females (detailed below). *Nrp2⁻/⁻* mice were crossed with *Nrp2^F/F* mice so that all embryos are *Nrp2^F/-*.

## Neuron labeling in vivo
### In utero electroporation

Performed on E13.5 *Nrp2^F/-* embryos of timed-pregnant females according to the published protocol (*Saito, 2006*). Briefly, anesthesia was delivered with intraperitoneal injection of sodium pentobarbital and, upon anesthetization, a small vertical incision was made on the abdominal cavity and embryos were gently pulled out. DNA was microinjected in the lateral ventricle with a fine and polished glass capillary tube placed in a mouth-controlled pipette, followed by the administration of five electric pulses of 30–35 V and 50 ms each with inter-pulse interval 950 ms, by the use of forceps-type electrodes. Embryos were placed back in the mother's abdomen, the incision was closed with silk sutures

or surgical staples and the mouse was placed back in the cage on a slide warmer, with hydrogel next to it, and monitored until full recovery. After the surgery and until pup delivery the pregnant female was kept in the High Risk Mouse Room. Newborn mice were sacrificed on postnatal day 28 (P28) and brains were examined for direct fluorescence with a stereoscope. Electroporated brains were processed for immunofluorescence that included triple staining for EGFP, dsRED, and Ctip2; the latter to confirm that labeled neurons used for phenotypic analysis are Ctip2$^+$ (layer V pyramidal neurons).

### Genetic labeling

The *Thy1-EGFP-m* line was crossed to *Zdhhc15*-KO mice for labeling of layer V cortical pyramidal neurons. *Zdhhc15$^{+/-}$;Thy1-EGFP* female mice were crossed to generate sex-matched wild-type and *Zdhhc15*-KO littermates expressing Thy1-EGFP. Thy1-EGFP$^+$ brains were subjected to double staining against EGFP to visualize neuronal architecture, and Ctip2 to confirm that EGFP-labeled neurons used for phenotypic analysis are Ctip2$^+$.

### Golgi staining

Performed with the use of Rapid GolgiStain kit (FD Neurotechnologies, Cat no. PK401) according to the manufacturer's protocol. Briefly, brains were dissected and incubated with impregnation solution A+B for 12 days in darkness at room temperature (solution was replaced by fresh the second day). After 12 days, brains were incubated with solution C for 4 days in darkness at 4°C (solution C was replaced by fresh the second day). Brains were embedded in NEG-50 frozen section medium following regular procedures and sectioned with a cryostat (Leica) at the sagittal plane at 100 μm thickness. Sections were transferred on gelatin-coated superfrost plus slides, left to dry in the dark overnight and next day they were stained according to the protocol. Neurons were imaged with acquisition of stacks using a DIC at an inverted LSM700 confocal microscope (Zeiss), with a 20× lens for visualization of dendrites and a 63× oil lens for visualization of dendritic spines.

## Phenotypic analysis

### Dendritic spine assay and analysis

For dendritic spine assessment in vitro, neurons were cultured on glass coverslips until DIV21 to allow spine formation. Between DIV7 and DIV9 they were transfected with pCIG2-ires-EGFP-expressing plasmids and at DIV21 they were treated with 5 nM Sema3F-AP or 5 nM AP (control) in neuron growth medium for 6 hr in the tissue culture incubator (37°C, 5% $CO_2$). Subsequently, neurons were fixed with ice-cold 4% paraformaldehyde for 10 min at room temperature and subsequently subjected to EGFP immunofluorescence. For both in vitro and in vivo analysis, EGFP-expressing neurons were imaged with an upright laser scanning confocal microscope (LSM700, Zeiss) with a 63× oil lens by the acquisition of Z-stacks. Imaging parameters were kept constant during image acquisition across all samples. For spine analysis, the 3D projection was calculated and spines were counted along the proximal 50 μm of the apical dendrite (relative to the cell body). All subtypes of dendritic spines (stubby, mushroom, filopodia-like) were counted.

### Dendritic arborization assay and analysis

For dendritic arborization assessment in vitro, neurons were cultured on glass coverslips until DIV12 to allow dendritic elaboration. At DIV12 primary cortical neurons were treated with 5 nM Sema3A-AP or 5 nM AP (control) for 6 hr in neuron growth medium in the tissue culture incubator (37°C, 5% $CO_2$). After treatment, they were fixed with 4% paraformaldehyde and subjected to immunofluorescence with an antibody directed against the somatodendritic marker MAP2. For both in vitro and in vivo analysis, imaging was performed at an LSM700 upright confocal microscope (Zeiss) with a 40× oil lens with the acquisition of Z-stacks. Imaging parameters were kept constant during image acquisition across all samples. The calculated 3D projection was thresholded and any background fluorescence or neighboring neurons interfering with the analysis were erased with the paintbrush tool. The point selection was placed at the center of the cell body and Sholl analysis (ImageJ) was run to count the dendritic processes at various distances from the cell body, ranging from 0 to 100 μm or 150 μm.

## Statistical analysis and software

Statistical significance and the size of samples analyzed are presented in figures and figure legends. Image analysis was performed using ImageJ software with appropriate plugins (Rasband, WS, ImageJ, U.S. National Institutes of Health, Bethesda, MD, USA, https://imagej.nih.gov/ij/, 1997–2016). The prediction of neuropilin palmitoylation sites was performed with the CSS-Palm 3.0 software, which is freely available (*Ren et al., 2008*). Conservation analysis of Nrp2 and Nrp1 amino acid sequences across species was performed with the ClustalW software. Details for the analysis of individual biochemical and phenotypic experiments are provided in separate sections. Data were collected in Excel and GraphPad Prism. Generation of all graphs and statistical analyses were performed in GraphPad Prism. The GraphPad Prism software was purchased. Figure assembly and generation of schematics presented in *Figure 7E and F* have been performed in Adobe Illustrator. Statistical analysis tests include two-tailed t test and analysis of variance (ANOVA) followed by the appropriate post hoc test for multiple comparisons (as mentioned in figure legends). The choice of post hoc statistical tests was based on the appropriate comparison-biological question under study in each experiment. For example, when one condition is used as a control and all other conditions are compared with the control, which is the case in many of our experiments, Dunnett's post hoc test is performed since it is the most appropriate test in these cases. For comparison of predefined pairs, such as in *Figure 4C* where each Nrp2 protein is separately examined for its ability to respond to Sema3F, Sidak's post hoc analysis is used since it assumes that each comparison is independent of the others, which is the case in this experiment. For comparison of every mean with every other mean, as it is the case in *Figure 5C*, Tukey's test is the most appropriate. Overall, besides selecting post hoc tests based on generally accepted determinations of appropriate statistical tests for the different experimental paradigms in this study, our post hoc test selection was also guided by GraphPad Prism, which provides, based on the particular biological question under study (i.e. sets of comparisons), recommended and non-recommended post hoc analysis methods. Statistical significance is defined as: *$p<0.05$; **$p<0.01$, ***$p<0.001$, ****$p<0.0001$, ns: not significant. Exact p values are presented either on graphs or in figure legends.

## Acknowledgements

We thank Dontais Johnson and Sarah Mitchell for excellent technical assistance. We also thank Martin M Riccomagno and other Kolodkin laboratory members for discussions. We thank Dr Joseph A Gogos for providing the *Zdhhc8*[-/-] mouse line and Dr Masaki Fukata for providing DHHC expression plasmids. This work was supported by NIH grant MH100024-Project #3 (ALK); previous support from the Howard Hughes Medical Institute (ALK); NIH grant R21NS085358 (EM, TW): and the State Scholarships Foundation (EK) and the AG Leventis Foundation (EK). DDG is an Investigator of the Howard Hughes Medical Institute.

## Additional information

### Competing interests

Randal Hand: Randal Hand is affiliated with Prilenia Therapeutics Development LTD. The author has no financial interests to declare. David D Ginty: Reviewing editor, *eLife*. The other authors declare that no competing interests exist.

### Funding

| Funder | Grant reference number | Author |
| --- | --- | --- |
| National Institutes of Health | MH100024-Project #3 | Alex L Kolodkin |
| State Scholarships Foundation | | Eleftheria Koropouli |
| A.G. Leventis Foundation | | Eleftheria Koropouli |

| Funder | Grant reference number | Author |
| --- | --- | --- |
| Howard Hughes Medical Institute | Previous support | Alex L Kolodkin |
| Howard Hughes Medical Institute | | David D Ginty |
| National Institutes of Health | R21NS085358 | Rebeca Mejías<br>Tao Wang |

The funders had no role in study design, data collection and interpretation, or the decision to submit the work for publication.

## Author contributions

Eleftheria Koropouli, Conceptualization, Formal analysis, Funding acquisition, Validation, Investigation, Methodology, Writing – original draft, Writing – review and editing; Qiang Wang, Randal Hand, Formal analysis, Validation, Investigation, Writing – review and editing; Rebeca Mejías, Investigation, Writing – review and editing; Tao Wang, Supervision, Writing – review and editing; David D Ginty, Conceptualization, Supervision, Funding acquisition, Writing – review and editing; Alex L Kolodkin, Conceptualization, Supervision, Funding acquisition, Validation, Methodology, Writing – original draft, Writing – review and editing

## Author ORCIDs

Eleftheria Koropouli (iD) http://orcid.org/0000-0001-8537-4471
Rebeca Mejías (iD) http://orcid.org/0000-0003-1936-7219
David D Ginty (iD) http://orcid.org/0000-0001-9723-8530
Alex L Kolodkin (iD) http://orcid.org/0000-0001-7562-5513

## Ethics

Animal procedures were carried out in conformity with the policies and guidelines of the Animal Care and Use Committee (ACUC) of the Johns Hopkins University (protocol # MO20M48), which are established according to the US National Research Council's Guide to the Care and Use of Laboratory Animals and in compliance with the Animal Welfare Act and Public Health Service Policy. Mice were handled with care and every effort was made to minimize suffering.

## Decision letter and Author response

Decision letter https://doi.org/10.7554/eLife.83217.sa1
Author response https://doi.org/10.7554/eLife.83217.sa2

# Additional files

## Supplementary files
• MDAR checklist

## Data availability

All data quantifications, western blots, and images that are presented in this study and for which links are provided in the manuscript as available on the eLife web site, in addition to all data relating to every graph, western blot and all quantifications, have been uploaded in order to be made easily available. Long-term archiving of data, as described above, that are part of this published study in eLife, along with all supporting data that contributed to each and every published file, will be managed by Johns Hopkins Data Services (JHUDS) using the Johns Hopkins Research Data Repository. This repository provides public access to data through an established platform supported by storage and preservation practices that follow the Open Archival Information System reference model. Deposited data has been given standard data citations and persistent identifiers (DOIs). Data are archived under a memorandum of understanding renewed every 5 years with the PI's consent. The DOI for these data is https://doi.org/10.7281/T1/OY2X8T.

The following dataset was generated:

| Author(s) | Year | Dataset title | Dataset URL | Database and Identifier |
|---|---|---|---|---|
| Koropouli E, Wang Q, Mejías R, Hand R, Wang T, Ginty DD, Kolodkin AL | 2023 | Data associated with the publication: Palmitoylation regulates neuropilin-2 localization and function in cortical neurons and conveys specificity to semaphorin signaling via palmitoyl acyltransferases | https://doi.org/10.7281/T1/OY2X8T | Johns Hopkins Research Data Repository, 10.7281/T1/OY2X8T |

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
