## [Editor Report]

Signaling mediated by Semaphorins and their receptors Nrp1 and Nrp2 is crucial for regulating the morphology of dendritic spines and dendritic arborization during development. In this manuscript, the authors found that the post-translational modification of S-palmitoylation dictates the subcellular localization and trafficking of Nrp2, but not Nrp1, and is required for Sema3F-dependent pruning of spines on the apical dendrites of layer V cortical neurons. The study provides important insights into how semaphorin signaling achieves spatial specificity on diverse downstream cellular events.

---

## [Decision Letter]

**Decision letter after peer review:**

Thank you for submitting your article "Palmitoylation regulates neuropilin-2 localization and function in cortical neurons and conveys specificity to semaphorin signaling via palmitoyl acyltransferases" for consideration by *eLife*. Your article has been reviewed by 3 peer reviewers, and the evaluation has been overseen by a Reviewing Editor and Catherine Dulac as the Senior Editor. The reviewers have opted to remain anonymous.

Essential revisions:

1) Please address concerns about extremely low N or N not stated for some experiments. This includes the following figures:

Figure 6A-C

Figure 6G-I (number of biological replicates not stated)

Figure 7 C-D

*Reviewer #1 (Recommendations for the authors):*

One major weakness for this study is the very low n number in several of the key experiments, indicated in the concerns (#1) in the Public Review. It is surprising that statistical analysis could even be performed as these quantitative data only have n=2 for independent experiment/animals per genotype: such as Figure 2E, 2G, Figure 3-Sup Figure 4E and Figure 3-Sup Figure 4G, Figure 4F, Figure 6C, and Figure 7D. While the authors count the number of cells/neurons measured as the n number in many of the quantifications throughout the paper, the number of "independent culture experiments" or the "number of animals per genotype" should really be used for the n number, and within each experiment/animal the (average or total) number of cells/neurons measures should be reported.

It is also standard to first perform a priori power analysis to determine the sample size for statistical significance. For example, the in vivo genetic interaction experiment between Nrp2 and DHHC15 (Figure 7C and 7D), only two brains per genotype were analyzed, while the number of neurons analyzed was used as the n number it is also well known that genetically modified animals can have a wide range of variability.

Also, why is several (at least 3-4) different type of multiple comparison tests being performed for similar quantifications? For example, in Figure 4B and 4C are measuring the number of spines per micrometer on dendrites of primary neurons transfected with the different cysteine mutant constructs with or without Sema3F treatment, but in one analysis (4B) the Bonferroni's test was used as the post-hoc while in the other (4C) the Sidak's test was used, and in Figure 4F the Dunnett's test was used, even when the n number was only 2 for the Sema3A treatment condition. At the very least, the authors should provide a rationale/justification in the Methods for the use of the different tests.

*Reviewer #2 (Recommendations for the authors):*

My main criticism is the N seems to be low (n=2) in several of the experiments, most importantly in the genetic interaction of Nrp2 and DHHC15, which is a highly important experiment for this paper.

*Reviewer #3 (Recommendations for the authors):*

1. In the methods, please specify whether or not microscope settings such as laser power was kept constant during data acquisition within an experiment. Please also specify when you are using stacks vs. single planes for your data analysis.

2. In Figure 3C, D-please clarify in the legend that this is total staining for Nrp-2 with anti-GFP antibody.

3. In Figure 7 suppl 1E there is no quantification of commissure. Although that level of detail is not critical for the major conclusions of this paper, perhaps you might comment on what happens to the commissure in Nrp2-/- mice?

4. I found this sentence confusing:

334 Given the surface mislocalization of palmitoylation-deficient Nrp-2, we next asked

335 how palmitoylation might affect Nrp-2 surface protein localization

Perhaps you meant "…we next asked how palmitoylation might affect Nrp-2 localization in Golgi?"

---

## [Author Response]

Essential revisions:1) Please address concerns about extremely low N or N not stated for some experiments. This includes the following figures:Figure 6A-C

For these ABE experiments in wild-type vs. *ZDHHC15*-KO primary cortical neurons, which include checking for intraexperimental reproducibility and control for potential differences in protein expression in input samples (endogenous protein), there are indeed 3 independent ABE assays for the *Nrp-2* condition, each with 3 biological replicas (neurons harvested from different animals/cultures) for each genotype, as shown in Figure 6A (I, II, III denote biological replicas). For Nrp-1, these data come from 2 independent ABE experiments, however, each experiment includes 3 biological replicas per genotype. This may have gone unnoticed but was included in Figure 6 legend; we clarify this further in our revised manuscript (lines 1697-1701).

We think that these results, owing to the number of replicas/experiment and the number of ABE assays, along with our observations that Sema3F/Nrp-2 signaling in *ZDHHC15*-KO neurons in vitro is affected (Figure 6D-F), whereas Sema3A/Nrp-1 is not (Figure 6G–I), together provide support for ZDHHC15 having an effect on Nrp-2 palmitoylation but no effect on Nrp-1 palmitoylation.

Figure 6G-I (number of biological replicates not stated)

For this experiment there are 3 different experiments, each of which employed an independent mouse of the indicated genotype—this is described in the Figure legend.

Figure 7 C-D

This is indeed the most important point raised, in our view, and it was raised by both reviewers 1 and 2. The reviewers are correct—there are only 2 brains per genotype. Though it is no excuse, the reason behind this very low number of animals is that for our quantifications we only considered mice from the same litter. To obtain all four alleles (*Nrp-2^WT^*, *Nrp-2 knockout*, *ZDHHC15^WT^*, and Z*DHHC15 knockout*) in the three different configurations along with Thy1-GFP expression (genetic labeling of layer V neurons) within the same litter required many, many matings. This is further complicated by the fact that the *ZDHHC15* gene is located at the X chromosome – as clarified in the revised manuscript (lines 383-384) – which necessitates the study of genes’ dose-dependent effects in heterozygous females. Given all these limitations, in the end, these were the only animals/neurons we obtained. Though these are all littermate controls, which is a strength, we realize that the animal numbers are low.

One possibility is not to include this experiment in our manuscript. However, as suggested by the Reviewing Editor, we provide to the reviewers in Author response image 1 additional graphs that break down these data for each of the two brains used for each of the three genotypes in this experiment, showing that there is no apparent difference in the distribution of data points between any 2 brains per genotype in these littermate pups.

**Author response image 1. sa2fig1:** 

However, we agree that our conclusions referring to this experiment, as suggested, must be softened, and we have moved this experiment from Figure 7C-D to Figure 7—figure supplement 1CD, indicating in the main text (lines 459-473) and supplemental figure legend the caveats associated with this experiment.We would like to point out that these experiments were performed some time ago, prior to the onset of the pandemic, and owing to issues relating to Covid 19-restrictions on experimentation and animal husbandry, we no longer have the mutants in our collection necessary for replication of these experiments to expand n’s. However, even if we did, it would take 6-9 months, at least, to obtain these same allele combinations, so in the end we are not able to increase the n’s for this experiment.

One could argue that genetic interactions of this sort are helpful but of course not definitive evidence of direct interactions between signaling pathway components, though over the years we have used this sort of argument in our work. In our current study there are also caveats associated with this sort of genetic interaction experiment, including that there are many other potential ZDHHC15 substrates besides neuropilin-2 (Nrp-2), and also that we cannot rule out with these data in hand, even if the n’s were substantially higher, haploinsufficiency issues in these genetic combinations which could mask the phenotypic effect of the tested genetic interaction (as shown previously for ZDHHC8)(Mukai et al., 2015, 2008, 2004). Moreover, given the location of *ZDHHC15* gene on the X chromosome, there could be phenotypic variability in *ZDHHC15* heterozygous females depending on random X-inactivation (lyonization).

We would also like to mention that in addition to this in vivo genetic interaction experiment, which suggests a direct interaction between Nrp-2 and ZDHHC15, there are other results in our manuscript that support a direct interaction between Nrp-2 and ZDHHC15 in cortical neurons:

1-Palmitoylation of Nrp-2 is reduced in *ZDHHC15*-KO cortical neurons in vitro

*2-ZDHHC15*-KO neurons in vitro do not respond to Sema3F, but do respond to Sema3A

3*-ZDHHC15*-KO neurons in vivo do show the same dendritic spine phenotype as it is observed in *Nrp-2* mutants (and we present here n = 4 mice for use of EGFP to mark neuronal morphology, and further an additional n = 3 mice using Golgi staining). Note that in Figure 7A-B, we state n = 4 mice per genotype (WT vs *ZDHHC15*-KO with Thy1-GFP). This was a mistake on our part (in our original submission we stated n = 3 mice/genotype), and it has been corrected in the revised manuscript. All source data available is provided, as per *eLife* publication policy.

Reviewer #1 (Recommendations for the authors):One major weakness for this study is the very low n number in several of the key experiments, indicated in the concerns (#1) in the Public Review. It is surprising that statistical analysis could even be performed as these quantitative data only have n=2 for independent experiment/animals per genotype: such as Figure 2E, 2G, Figure 3-Sup Figure 4E and Figure 3-Sup Figure 4G, Figure 4F, Figure 6C, and Figure 7D. While the authors count the number of cells/neurons measured as the n number in many of the quantifications throughout the paper, the number of "independent culture experiments" or the "number of animals per genotype" should really be used for the n number, and within each experiment/animal the (average or total) number of cells/neurons measures should be reported.

We agree with the reviewer regarding the effects of individual cysteine residues on Nrp-2 palmitoylation, localization and dendritic spine pruning. These observations lead us to speculate about potential mechanisms that regulate Nrp-2 palmitoylation and function. Based on our experiments, it appears that the effect of mutating cysteine residues is not strictly additive; there seems to be a predominant role of select cysteine residues, e.g. C878, in regulating overall Nrp-2 palmitoylation. Similarly, single cysteine residues appear to be required for dendritic spine pruning to variable extents. The prevailing mechanism seems to be a dominant/synergistic effect among cysteine residues within the same cysteine cluster, such that when one cysteine residue is mutated there is a disproportional change in protein palmitoylation and function.

However, in our protein clustering assay, we did observe that the effect of cysteine mutations on Nrp-2 localization is roughly additive. Specifically, single cysteine mutants did not have any detectable deficit in our clustering assay (*data not shown*), whereas the TCS mutant invariably exhibited diffuse surface localization in both neurons and heterologous cell lines. We think that loss of clustering is a major deficit and assays that can reveal more specific effects on protein localization should be part of future studies in order to explain functional deficits of single cysteine mutants in the case of Nrp-2 and other palmitoylated proteins.

We have addressed this excellent point raised by this reviewer in our Discussion to explain these observations and models for Nrp-2 palmitoylation (lines 547-561).

It is also standard to first perform a priori power analysis to determine the sample size for statistical significance. For example, the in vivo genetic interaction experiment between Nrp2 and DHHC15 (Figure 7C and 7D), only two brains per genotype were analyzed, while the number of neurons analyzed was used as the n number it is also well known that genetically modified animals can have a wide range of variability.

Regarding the genetic interaction experiment, Figure 7C-D, please see the explanation we provide in point #3 of the “Essential Revisions” described above.

Also, why is several (at least 3-4) different type of multiple comparison tests being performed for similar quantifications? For example, in Figure 4B and 4C are measuring the number of spines per micrometer on dendrites of primary neurons transfected with the different cysteine mutant constructs with or without Sema3F treatment, but in one analysis (4B) the Bonferroni's test was used as the post-hoc while in the other (4C) the Sidak's test was used, and in Figure 4F the Dunnett's test was used, even when the n number was only 2 for the Sema3A treatment condition. At the very least, the authors should provide a rationale/justification in the Methods for the use of the different tests.

When we perform ANOVA and observe statistical significance differences, a post hoc (multiple comparisons) test should be conducted to examine which groups are different from each other. There are several different post hoc tests, each appropriate for a different type of comparison. Therefore the appropriate post hoc test was chosen based on the desired comparison and the biological question being addressed; this is why we have used different tests for different experiments in this study. For example, when different categories are compared with a single control, as it is the case in many of our experiments, Dunnett’s post hoc test is the most appropriate; for comparison of predefined pairs, Bonferroni or Sidak’s are usually used. Moreover, in several cases two different post-hoc tests have been performed in order to look for consistency of the results regardless of the test used. For example, in Figure 4B, Dunnett’s and Bonferroni’s methods give the same degree of statistical significance; the same is true in Figure 4C with Sidak’s and Bonferroni’s methods. To clarify how we were guided in our choice of post hoc tests following ANOVA, and also all of the other statistical tests we use in this study, we have added a rationale statement in the Materials and methods–Statistical analysis and software subsection (lines 13201336).

Reviewer #2 (Recommendations for the authors):My main criticism is the N seems to be low (n=2) in several of the experiments, most importantly in the genetic interaction of Nrp2 and DHHC15, which is a highly important experiment for this paper.

The main criticism by this reviewer regarding the low Ns in the genetic interaction experiment between *Nrp-2* and *ZDHHC15*, presented in our former Figure 7C-D, is addressed in point #3 of the Essential Revisions presented above.

Reviewer #3 (Recommendations for the authors):1. In the methods, please specify whether or not microscope settings such as laser power was kept constant during data acquisition within an experiment. Please also specify when you are using stacks vs. single planes for your data analysis.

The imaging parameters were kept constant during data acquisition in each experiment and also across all experimental replicas. Additional statements have been added in the “Materials and methods” section for each experiment to provide clarity on this important issue.

2. In Figure 3C, D-please clarify in the legend that this is total staining for Nrp-2 with anti-GFP antibody.

A clarification statement has been added in the legend of Figure 3C-D, that this is indeed total staining for Nrp-2 protein with an antibody directed against GFP, and it is also explained in the Materials and methods (lines 1145-1146).

3. In Figure 7 suppl 1E there is no quantification of commissure. Although that level of detail is not critical for the major conclusions of this paper, perhaps you might comment on what happens to the commissure in Nrp2-/- mice?

The reviewer is correct that we have not quantified the size of the anterior commissure (AC) in the *ZDHHC15*-KO mutants we analyzed. In *Nrp-2^-/-^* mice, the AC is barely detectable or totally absent (Giger et al., 2000). So, in our current assessment, we evaluated the presence and overall appearance of the anterior commissure in mutant mice as compared to wild-type mice. Since visual inspection did not reveal any differences similar to what we have presented previously regarding AC morphology in *Nrp-2* mutants, we think it is very unlikely that AC has any deficits in *ZDHHC15* mutant mice that are similar to those observed in *Nrp-2* mutants; this is why we did not examine this in great detail beyond our visual assessment of overall AC morphology.

4. I found this sentence confusing:334 Given the surface mislocalization of palmitoylation-deficient Nrp-2, we next asked335 how palmitoylation might affect Nrp-2 surface protein localizationPerhaps you meant "…we next asked how palmitoylation might affect Nrp-2 localization in Golgi?"

The sentence “Given the surface mislocalization of palmitoylation-deficient Nrp-2, we next asked how palmitoylation might affect Nrp-2 surface protein localization” has been rephrased for clarity as suggested by this reviewer:

“Given the surface mislocalization of palmitoylation-deficient Nrp-2, we next addressed the early intracellular trafficking defects that lead to aberrant Nrp-2 delivery to the cell surface.”.

References

Giger RJ, Cloutier JF, Sahay A, Prinjha RK, Levengood D V., Moore SE, Pickering S, Simmons D, Rastan S, Walsh FS, Kolodkin AL, Ginty DD, Geppert M. 2000. Neuropilin-2 is required in vivo for selective axon guidance responses to secreted semaphorins. *Neuron* 25:29–41. doi:10.1016/S0896-6273(00)80869-7

Mukai J, Dhilla A, Drew LJ, Stark KL, Cao L, MacDermott AB, Karayiorgou M, Gogos JA. 2008. Palmitoylation-dependent neurodevelopmental deficits in a mouse model of 22q11 microdeletion. *Nat Neurosci* 11:1302–1310. doi:10.1038/nn.2204

Mukai J, Liu H, Burt RA, Swor DE, Lai W-S, Karayiorgou M, Gogos JA. 2004. Evidence that

the gene encoding ZDHHC8 contributes to the risk of schizophrenia. *Nat Genet*
**36**:725–

731. doi:10.1038/ng1375

Mukai J, Tamura M, Fénelon K, Rosen AM, Spellman TJ, Kang R, MacDermott AB, Karayiorgou M, Gordon JA, Gogos JA. 2015. Molecular substrates of altered axonal growth and brain connectivity in a mouse model of schizophrenia. *Neuron*
**86**:680–95. doi:10.1016/j.neuron.2015.04.003